# A systematic review and meta-analysis of adolescent nutrition in Ethiopia: Transforming adolescent lives through nutrition (TALENT) initiative

**Mubarek Abera**[1]*, **Abdulhalik Workicho**[2], **Melkamu Berhane**[1], **Desta Hiko**[2], **Rahma Ali**[2], **Beakal Zinab**[2], **Abraham Haileamlak**[1], **Caroline Fall**[3]

**1** Faculty of Medical Sciences, Institute of Health, Jimma University, Jimma, Ethiopia, **2** Faculty of Public Health, Institute of Health, Jimma University, Jimma, Ethiopia, **3** Medical Research Council (MRC) Lifecourse Epidemiology Centre, Southampton General Hospital, University of Southampton, Southampton, United Kingdom

* Mubarek.abera@gmail.com

**Data Availability Statement:** All relevant data are within the manuscript and its Supporting Information files.

## Abstract

### Background

Ethiopia has undergone rapid economic growth over the last two decades that could influence the diets and nutrition of young people. This work systematically reviewed primary studies on adolescent nutrition from Ethiopia, to inform future interventions to guide policies and programs for this age group.

### Method

A systematic search of electronic databases for published studies on the prevalence of and interventions for adolescent malnutrition in Ethiopia in the English language since the year 2000 was performed using a three-step search strategy. The results were checked for quality using the Joanna Bridge Institute (JBI) checklist, and synthesized and presented as a narrative description.

### Results

Seventy six articles and two national surveys were reviewed. These documented nutritional status in terms of anthropometry, micronutrient status, dietary diversity, food-insecurity, and eating habits. In the meta-analysis the pooled prevalence of stunting, thinness and overweight/obesity was 22.4% (95% CI: 18.9, 25.9), 17.7% (95% CI: 14.6, 20.8) and 10.6% (7.9, 13.3), respectively. The prevalence of undernutrition ranged from 4% to 54% for stunting and from 5% to 29% for thinness. Overweight/obesity ranged from 1% to 17%. Prevalence of stunting and thinness were higher in boys and rural adolescents, whereas overweight/obesity was higher in girls and urban adolescents. The prevalence of anemia ranged from 9% to 33%. Approximately 40%-52% of adolescents have iodine deficiency and associated

**Funding:** The study was funded by a Global Challenges Research Fund/Medical Research Council pump priming grant (grant number: MC_PC_MR/R018545/1). The funders had no involvement in the design, implementation, analysis or decision to prepare a manuscript or to publish the finding.

**Competing interests:** The authors have declared that no competing interests exist.

risk of goiter. Frequent micronutrient deficiencies are vitamin D (42%), zinc (38%), folate (15%), and vitamin A (6.3%).

## Conclusions

The adolescent population in Ethiopia is facing multiple micronutrient deficiencies and a double-burden of malnutrition, although undernutrition is predominant. The magnitude of nutritional problems varies by gender and setting. Context-relevant interventions are required to effectively improve the nutrition and health of adolescents in Ethiopia.

## Introduction

Adolescence, the transition from childhood to adulthood, is a period of rapid growth and development, second only to fetal life and infancy. About 20% and 40% of the final adult height and weight respectively are attained during this stage [1]. Physical growth (muscle and bone size and density), the onset of puberty and (in girls) menstruation, and major psychological, emotional and cognitive maturation, are vital changes taking place during adolescence, creating a greater demand for protein and energy [2], and a continued need for micronutrients [3]. Equally, the rapid mental, psychological and social development during adolescence increases the risk of psychosocial problems, [4] which may alter adolescents' nutrition behavior.

A 2018 United Nation Children's Fund (UNICEF) report showed that globally, adolescents constitute 16% of the population [5]. However, this figure in Ethiopia is 34% [6]. Traditionally, under nutrition has been the predominant public health nutrition problem in low income settings, while in high income countries overnutrition (overweight and obesity) was more prevalent. However, as many low- and middle-income countries (LMICs) are experiencing rapid economic growth, with concomitant urbanisation and lifestyle transformations, overweight and obesity are emerging as significant problems among the adolescent population. This, alongside persisting undernutrition in large sections of the population results in a double burden of malnutrition in LMICs [7], compounded by low levels of government investment to solve the problem [8, 9].

Malnutrition in LMICs often starts prenatally, continues through childhood and adolescence, and even extends to adulthood. This creates a vicious cycle of malnutrition contributing to adverse intergenerational effects, such as low birth weight, which in turn has lifelong effects on health [10]. Adolescence can be viewed as a period of opportunity in which catch-up growth can occur and when nutritional status can be optimized prior to parenthood [11]. Thus designing appropriate interventions to support safe, healthy and productive transition from childhood to adulthood is a critical step to end malnutrition and improve the overall health and wellbeing of society [12].

Although nutritional interventions are growing rapidly in Ethiopia, most of these are targeted to young children, and pregnant and lactating women, leaving adolescents a relatively neglected group. Many of the successes achieved in maternal and child health in Ethiopia over the last few decades are the result of specific investment in the health and nutrition of children and women. Though the 2017 revised National Nutrition Program (NNP) and the 2018 Food and Nutrition Policy (FNP) [13, 14] highlighted issues of adolescent nutrition, there is no equivalent investment in adolescents' distinctive health and nutrition needs in Ethiopia [15]. In order to inform this process, we have conducted this systematic review and meta-analysis of

original research from Ethiopia, thereby collating current evidence on nutritional status of Ethiopian adolescents and the effectiveness of nutritional interventions in this age group.

## Methods

### Search strategy

We adopted a rigorous systematic approach [16, 17]. The first step was to formulate objectives/research questions as follows.

1. From observational studies and surveys, what is known about the nutritional status of Ethiopian adolescents (boys and girls age 10–19 years) in terms of: a) body size and energy balance (chiefly weight, BMI and height), b) micronutrient status, c) dietary intake, diversity and quality, and d) dietary behaviours?

2. From their associations with population characteristics, what is known about the possible determinants of these aspects of nutritional status? Which adolescents are at risk of nutritional problems?

3. From intervention studies, what is known about the effectiveness of nutritional interventions in the Ethiopian population?

A three-step comprehensive literature search strategy was used to locate relevant literature published over the last 20 years from Ethiopia. Firstly, we set relevant key words and terms, using a logic grid for each key term. The terms used included "nutrition", "micronutrient", "malnutrition", "undernutrition", stunting", "thin", "obesity", "food insecurity" "dietary diversity" "anemia", "iron", "folic", "vitamin", "zinc", "iodine", "copper", "magnesium", "selenium" and "eating disorder". The terms we used to define the population were "adolescent", "teenage", "youth", "school children" and "young child", and setting in "Ethiopia".

### Data sources/base

The search query was first developed for PubMed and later extended to EBSCO/ERIC and EBSCO/CINAHL to identify different concepts in the literature. Secondly, we carried out the search, expanding all terms in specific databases. Thirdly, we manually searched the reference lists of the identified studies.

### Study selection process

Following the search, two researchers (AW & DH) screened studies by title. Then two independent researchers (BZ & RA) screened the abstracts and assessed the eligibility for full text retrieval. Selected full-text studies were compared between the reviewers, with disagreements being resolved through discussion and consensus with a 3rd researcher (MA).

### Population

Adolescent age 10–19

### Outcomes

Nutritional status measured by anthropometric indices: Stunting (Height-for-age z-score <-2) Thinness (BMI-for-age z-score <-2), underweight (weight-for-age z-score <-2), overweight (Body mass index-for-age z-score is >1), Obesity (Body mass index-for-age z-score is >2), combined overweight/obesity (Body mass index-for-age z-score is >1) micronutrient status, dietary diversity score (DDS) measured with the Food and Agriculture Organization of the

United Nations (FAO), and food insecurity score assessed with Household Food Insecurity Access Scale (HFIAS).

## Study selection criteria

**Inclusion criteria.** The inclusion criteria were developed through discussion in an iterative process. Primary studies or national government surveys (DHS and Micronutrient survey) involving human subjects, reporting quantitative outcomes, published in English between the year 2000 and 2020, conducted in Ethiopia, among adolescents aged 10–19 years were included. Therefore, in this review, studies which researched nutritional status including but not limited to under nutrition, over nutrition, micronutrient status or deficiency, food insecurity, diet diversity, dietary behavior, diet quality, eating disorders and protein energy deficiency among adolescents in Ethiopia were included. A PRISMA flow diagram is included to inform the study selection process.

**Exclusion criteria.** Unpublished studies, articles published in a language other than English, reviews, book reviews, commentaries, letters to the editor and case reports, publications with only an abstract, and studies conducted outside Ethiopia were excluded. Qualitative studies were excluded because these are reviewed elsewhere by the TALENT collaboration [18].

## Data charting and synthesis of the results

Data were extracted using a pre-tested two-step process. Firstly, we developed a template (authors, year, settings (urban/rural), methodology, study question, study design, population, outcomes, and study quality). Secondly, each reviewer independently extracted data, which was then compared and any discrepancies discussed and resolved. The study findings are synthesized using narrative descriptions based on individual indicators that emerged. Meta-analysis was done for stunting, thinness and overnutrition (overweight/obesity) using random effect model with restricted maximum likelihood (REML) method. We used STATA version 17 for the meta-analysis. Forest plot for proportion with 95% confidence interval (CI) was reported. A Preferred Reporting Items for Systematic reviews and Meta-Analyses Checklist was used to guide reporting [19].

## Quality assessment

The Joanna Briggs Institute (JBI) quality assessment checklists for observational [20] and interventional [21] studies were used and the quality of the reviewed studies rated as low, medium and high. Studies rated as low quality were excluded from the review.

## Results

### Study selection

The initial search strategy and additional manual search identified 4153 records of which 3220 were removed due to the age of the study population (1810), outcomes mixed with other non-relevant indicators (1131), year of publication (275), language (2) and duplicates (2), leaving 933 records (Fig 1). Of these, 25 studies were secondary reviews or meta-analyses, which resulted in 908 records for full text review. Through full text review, 830 records were excluded as they were not relevant to our objectives, leaving 78 studies. Then finally a total of 74 articles were used for data extraction. No studies were excluded because of poor quality. Our last search date was on 1, November 2022.

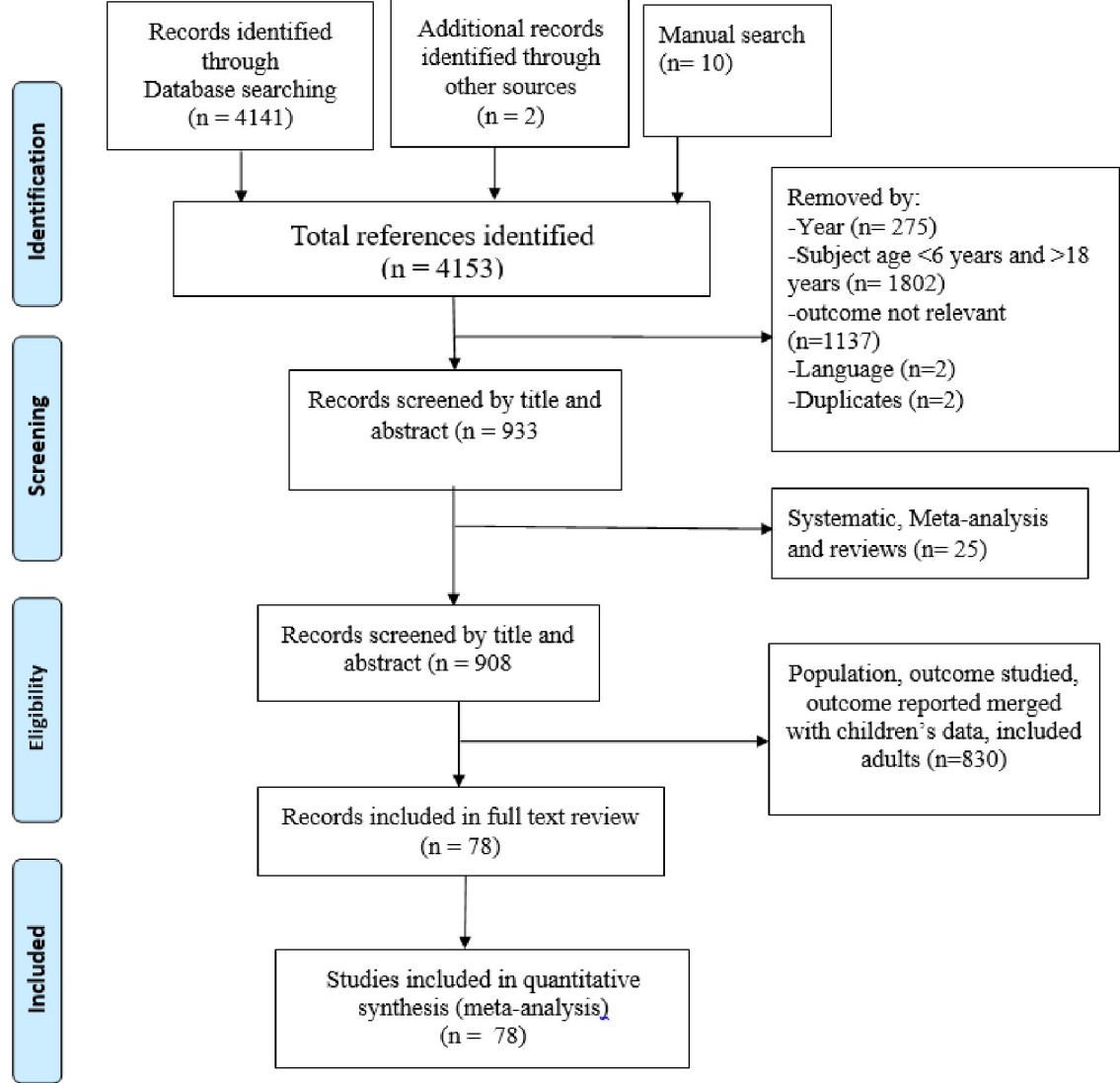

**Fig 1. Flow chart for search and selection process of articles on adolescent nutrition and health in Ethiopia.**

### Characteristics of included studies

The majority of the reviewed studies were observational and cross sectional; there were six longitudinal [22–27] and one interventional (quasi experimental) [28] studies. The outcomes assessed included body size and growth (41 studies) [22, 28–66], micronutrient status and deficiencies (15 studies) [39, 67–80], diet diversity (9 studies) [28, 44, 72, 80–85] and food insecurity (7 studies) [23–27, 46, 86]. There were two studies which addressed eating disorders [87, 88]. Ten studies reported multiple outcomes, and are described in more than one section of the review.

### Findings of the review

**Nutritional status defined by anthropometry, and its determinants.**   Forty studies, on a sample of 25 397 adolescents, and one demographic and health survey (DHS) [75] addressed nutritional status as defined by height, weight and BMI. Five specifically assessed over-

nutrition (overweight and/or obesity) [43, 44, 48, 50, 53], Eight assessed both undernutrition (stunting, thinness or underweight) and over-nutrition [29, 31, 32, 39–41, 45, 89] and 25 reported only undernutrition [22, 30, 33–38, 42, 46, 47, 49, 51, 52, 55–66]. The national adolescent nutrition surveys (DHS) reported stunting, thinness and overweight. Twenty one (including the DHS) were conducted in mixed urban and rural settings while thirteen and six studies were conducted only in urban and rural settings respectively. Two did not describe the study setting.

From studies that reported Z-scores for adolescents' height-for-age (HAZ) and body-mass-index (BMI)-for-age (BAZ) using the 2007 WHO growth reference, the minimum and maximum mean HAZ was -1.5 [33] and -0.5 [41] while they were -1.29 [33] and 0.44 [89] for BAZ respectively. A comparison between urban and rural adolescents showed that the mean BAZ and HAZ were significantly higher in urban than rural adolescents, with mean differences of 0.2 (95% confidence interval (CI): 0.02–0.34) and 0.58 (95% CI 0.45–0.72), respectively [32]. Both HAZ and BAZ, even for urban adolescents were, however, lower than the WHO reference data [32].

In the meta-analysis the pooled prevalence of stunting, thinness and overnutrition (overweight/obesity) were 22.4% (95% CI: 18.9, 25.9), 17.7% (95% CI: 14.6, 20.8) and 10.6% (7.9, 13.3), respectively as shown in Figs 2–4.

**Nutritional status by sex and setting.** Fig 5 shows the prevalence of stunting, thinness, underweight and overweight/obesity across the studies included, arrayed by year of publication; these outcomes are also shown stratified by sex and setting (rural/urban) in Figs 6–11. Among studies that measured the prevalence of undernutrition, the prevalence of stunting ranged from 4.4% in girls (urban 1.9% & rural 6.9%) in southwest Ethiopia [32] and 5.2% (5.9% boys, 4.4% girls, 4.2% urban & 8.8% rural) in Wolaita Sodo (south Ethiopia) [41] to 53.9% (urban 48.4% & rural 55.3%) in a sample of both sexes from northwest Ethiopia [30]. In terms of sex, stunting ranged from 5.9% in boys and 4.4% in girls in Wolaita Sodo town [41] and 7.2% in boys and 6.9% in girls (total 7.7%) in Addis Ababa city [31] to 47.4% in boys and 47.4% in girls in rural northwestern Ethiopia [57]. The prevalence of thinness ranged from 4.9% (boys 3.1%, girls 6.6%) in central Northern Ethiopia [59] to 44% (54.7% in boys and 33.7% in girls) in eastern Tigray (north Ethiopia) [36]. Thinness, in terms of sex, ranged from 3.1% [59] to 54.5% [36] in boys, and from 1.4% [51] to 48.4% [61] in girls. In terms of setting, stunting ranged from 1.9% in urban and 6.9% in rural settings [32] to 48.4% urban and 55.3% rural settings [30]. The prevalence of overweight/obesity ranged from 0.6% in girls (urban 1% & rural 0.3%) in southwestern Ethiopia [32] to 17% (boys 14% & girls 20%) in Addis Ababa city [50].

In ten of the fourteen studies that reported stunting in both sexes, the prevalence was higher in boys [22, 31, 33, 36, 41, 46, 49, 51, 55, 63] while in four stunting was higher in girls [38, 45, 59, 62]. Nine of twelve studies, reported more thinness in boys than girls [33, 36, 41, 45, 46, 51, 55, 62, 63]. In all of the five studies that have reported overnutrition, the prevalence of overweight/obesity was higher in girls [29, 31, 41, 43–45, 50, 53], and in three of four studies that have reported under nutrition, the prevalence of underweight was higher in boys [29, 31, 36].

**Nutritional status by age.** In eight of fifteen and in nine of thirteen studies, the prevalence of stunting and thinness was higher in younger (10–14 years) compared to older (15–18 years) adolescents. The prevalence of stunting ranged from 2.1% [47] to 38.5% [64] in younger (10–13 years), and from 3.5% [46] to 61.7% [47] in older (16–18 years) adolescents. Thinness in younger adolescents ranged from 11% in Jimma zone, southwestern Ethiopia [45] to 64.3% early (10–13 years) in northwest Ethiopia [56], and in older adolescents it ranged from 3.8% in southwestern Ethiopia (Wollega Zone) [55] to 39.7% in eastern Ethiopia [52].

**Nutritional status trends over time.** We searched for individual articles published since 2000, but data on the nutritional status of adolescents was only available between 2010 and 2022 The data from individual studies showed no clear pattern across time (Figs 2–8); and

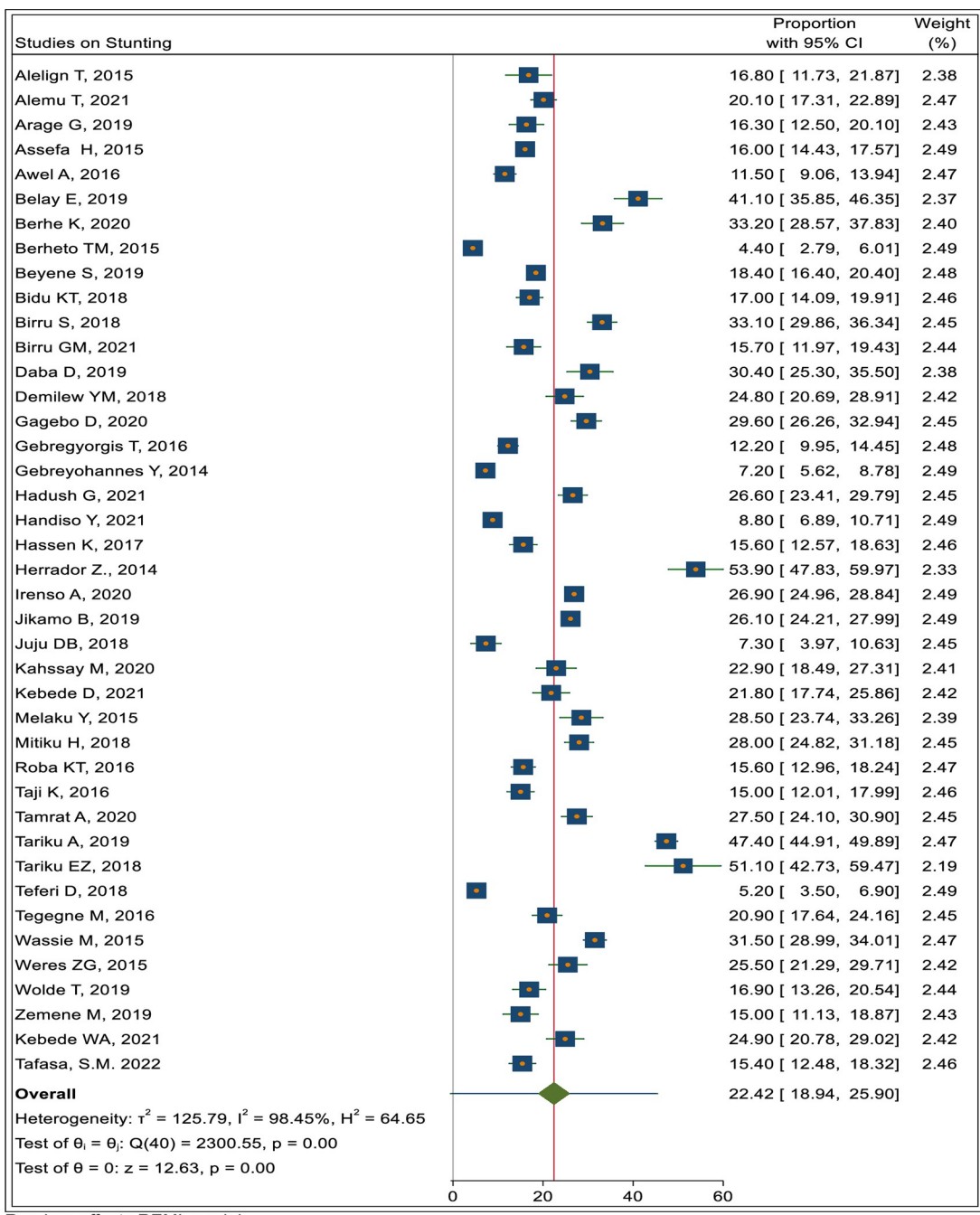

**Fig 2. Pooled prevalence of adolescent stunting in Ethiopia.**

both undernutrition and overnutrition have co-existed over the last 10 years. A high prevalence of stunting (54% and 51%) was documented in 2014 and 2018, respectively; while a prevalence of 80.8% underweight and 44% thinness was documented in 2015. Likewise, the highest prevalence of overweight (17%) was documented in 2018.

**Predictors of nutritional status.** In ten of thirteen studies, adolescents from rural areas were more likely to be stunted compared to urban adolescents [22, 30, 32, 33, 39, 41, 51, 59, 62,

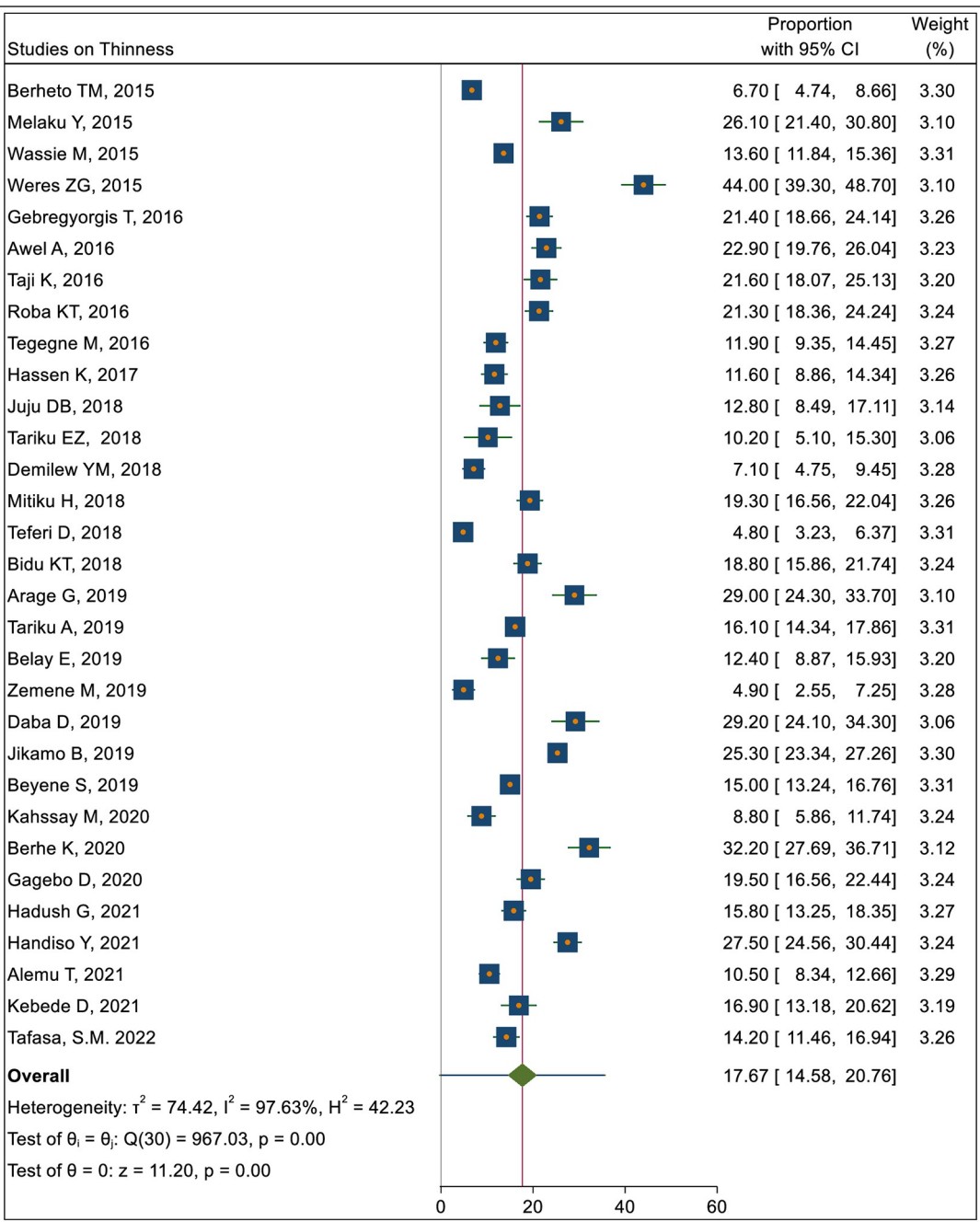

| Studies on Thinness | Proportion with 95% CI | Weight (%) |
|---|---|---|
| Berheto TM, 2015 | 6.70 [ 4.74, 8.66] | 3.30 |
| Melaku Y, 2015 | 26.10 [ 21.40, 30.80] | 3.10 |
| Wassie M, 2015 | 13.60 [ 11.84, 15.36] | 3.31 |
| Weres ZG, 2015 | 44.00 [ 39.30, 48.70] | 3.10 |
| Gebregyorgis T, 2016 | 21.40 [ 18.66, 24.14] | 3.26 |
| Awel A, 2016 | 22.90 [ 19.76, 26.04] | 3.23 |
| Taji K, 2016 | 21.60 [ 18.07, 25.13] | 3.20 |
| Roba KT, 2016 | 21.30 [ 18.36, 24.24] | 3.24 |
| Tegegne M, 2016 | 11.90 [ 9.35, 14.45] | 3.27 |
| Hassen K, 2017 | 11.60 [ 8.86, 14.34] | 3.26 |
| Juju DB, 2018 | 12.80 [ 8.49, 17.11] | 3.14 |
| Tariku EZ, 2018 | 10.20 [ 5.10, 15.30] | 3.06 |
| Demilew YM, 2018 | 7.10 [ 4.75, 9.45] | 3.28 |
| Mitiku H, 2018 | 19.30 [ 16.56, 22.04] | 3.26 |
| Teferi D, 2018 | 4.80 [ 3.23, 6.37] | 3.31 |
| Bidu KT, 2018 | 18.80 [ 15.86, 21.74] | 3.24 |
| Arage G, 2019 | 29.00 [ 24.30, 33.70] | 3.10 |
| Tariku A, 2019 | 16.10 [ 14.34, 17.86] | 3.31 |
| Belay E, 2019 | 12.40 [ 8.87, 15.93] | 3.20 |
| Zemene M, 2019 | 4.90 [ 2.55, 7.25] | 3.28 |
| Daba D, 2019 | 29.20 [ 24.10, 34.30] | 3.06 |
| Jikamo B, 2019 | 25.30 [ 23.34, 27.26] | 3.30 |
| Beyene S, 2019 | 15.00 [ 13.24, 16.76] | 3.31 |
| Kahssay M, 2020 | 8.80 [ 5.86, 11.74] | 3.24 |
| Berhe K, 2020 | 32.20 [ 27.69, 36.71] | 3.12 |
| Gagebo D, 2020 | 19.50 [ 16.56, 22.44] | 3.24 |
| Hadush G, 2021 | 15.80 [ 13.25, 18.35] | 3.27 |
| Handiso Y, 2021 | 27.50 [ 24.56, 30.44] | 3.24 |
| Alemu T, 2021 | 10.50 [ 8.34, 12.66] | 3.29 |
| Kebede D, 2021 | 16.90 [ 13.18, 20.62] | 3.19 |
| Tafasa, S.M. 2022 | 14.20 [ 11.46, 16.94] | 3.26 |
| **Overall** | 17.67 [ 14.58, 20.76] | |

Heterogeneity: $\tau^2 = 74.42$, $I^2 = 97.63\%$, $H^2 = 42.23$

Test of $\theta_i = \theta_j$: Q(30) = 967.03, p = 0.00

Test of $\theta = 0$: z = 11.20, p = 0.00

Random-effects REML model

**Fig 3. Pooled prevalence of adolescent thinness in Ethiopia.**

64]. The prevalence of thinness was higher among adolescents from less educated mothers, adolescents who have <3 meals per day and those from households comprising more than five people. In addition, adolescents who were physically inactive and adolescents with sedentary lifestyles were more likely to be obese than others [44].

The reviewed articles [22, 30, 32, 33, 39, 41, 51, 59, 62, 64] also identified common predictors for under- as well as overnutrition. Low dietary diversity, low frequency of daily food

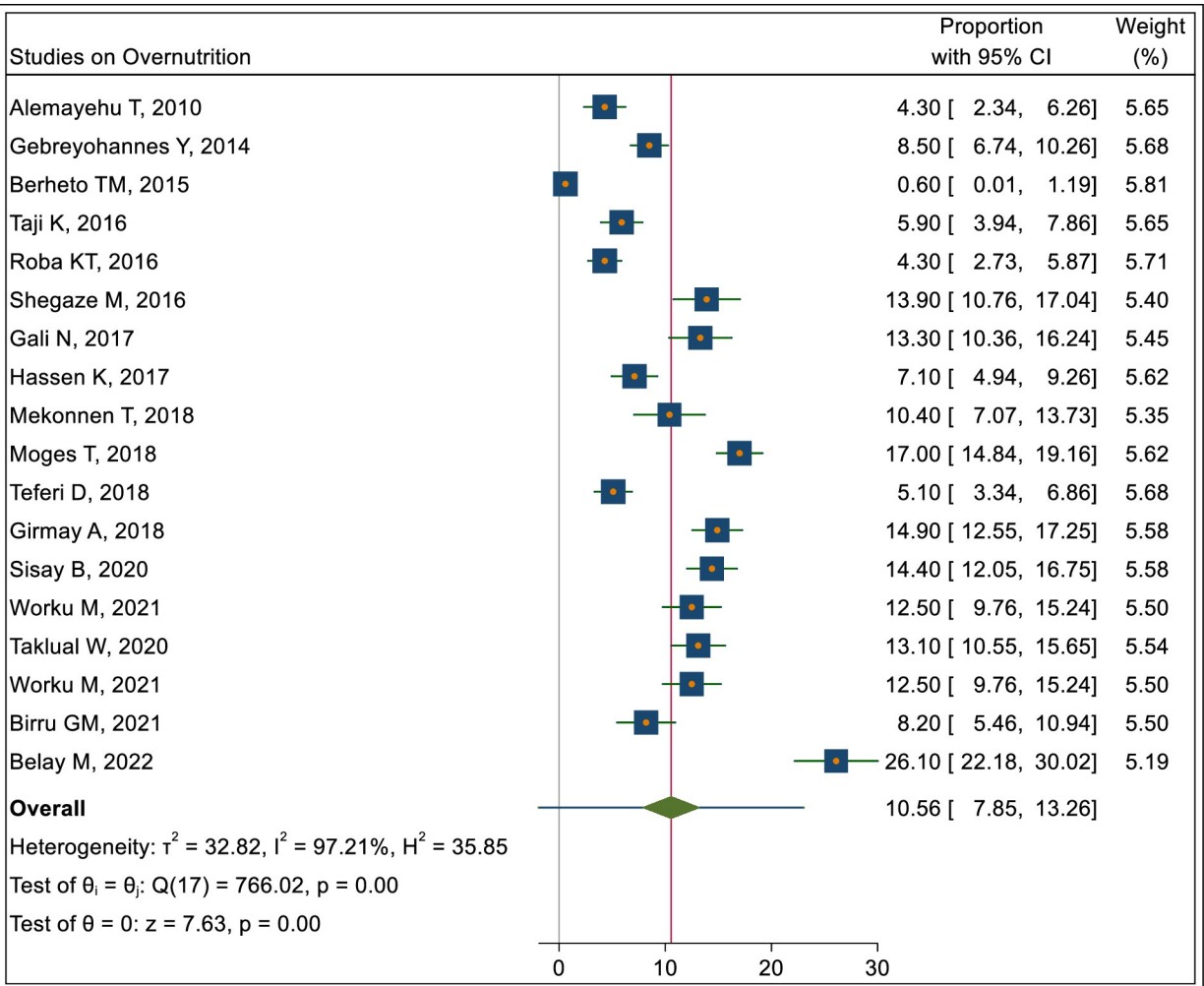

**Fig 4. Pooled prevalence of overnutrition (overweight/obesity) in Ethiopia.**

intake, higher household family size, low maternal education, food insecurity, and poor quality sources of drinking water were associated with undernutrition. In contrast, residency in urban settings, female sex, low levels of physical activity and a more sedentary life style are predictors of overweight/obesity (Table 1).

**Micronutrient deficiencies.** Thirteen studies [39, 67, 68, 70–74, 76–80], on a sample of 7 019 adolescents, and one national micronutrient survey [69] on a sample of 722, and the DHS [75] on adolescent nutrition reported micronutrient status. The national micronutrient survey reported a mean Vitamin A concentration (retinol) of 1.20 µmol/l and 6.3% with retinol <0.7 µmol/l in the age group 12–14 years; equivalent data for the 15–19 year age group were 1.40±0.43, and 3.2% [69]. In the same survey for the same age group, zinc deficiency was found in 38%, Vitamin B12 deficiency (<203 pg/ml) in 13.6%, severe iodine deficiency (<20 µg/L) in 1.9%, moderate iodine deficiency (50–99.9 µg/L) in 25.2% and mild iodine deficiency (20–49.9 µg/L) in 20.5%. Excess iodine (>300 µg/L) was 12.2% in 12–14 years age groups [69].

Based on the micronutrient survey, the national prevalence of anaemia was 14.9% (14.4% moderate (Hb 8–12 g/dL, 0.5% severe (Hb<8g/dL) in adolescents aged 12–14 years and 11.8%

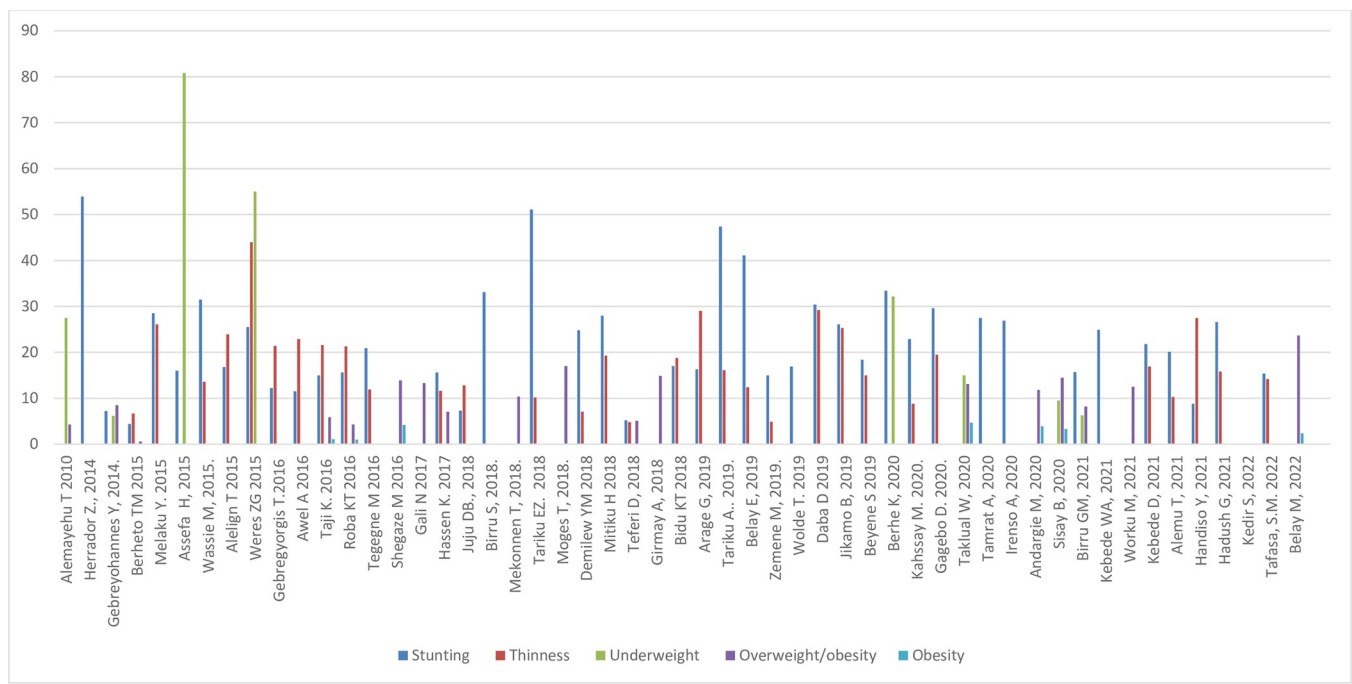

**Fig 5. Trends in the nutritional status of adolescent in Ethiopia over the last 10 years.**

(10.5% moderate and 1.3% severe) in the age group 15–19 years [69]. Iron deficiency (ferritin<15 μg/L) was 8.6% and iron deficiency anemia (IDA) was 2.6% in the age group 12–14; equivalent data for the 15–19 year age group were 10.0% and 3.2% [69].

Site specific studies in northwestern Ethiopia indicated a prevalence of anaemia of 13.4% (Hb < 12 g/dl [68] and 25.5% (92.4% mild, 5.9% moderate and 1.7% severe) with the odds of having anaemia higher in those with inadequate diet diversity score (DDS) (AOR = 2.1;95%

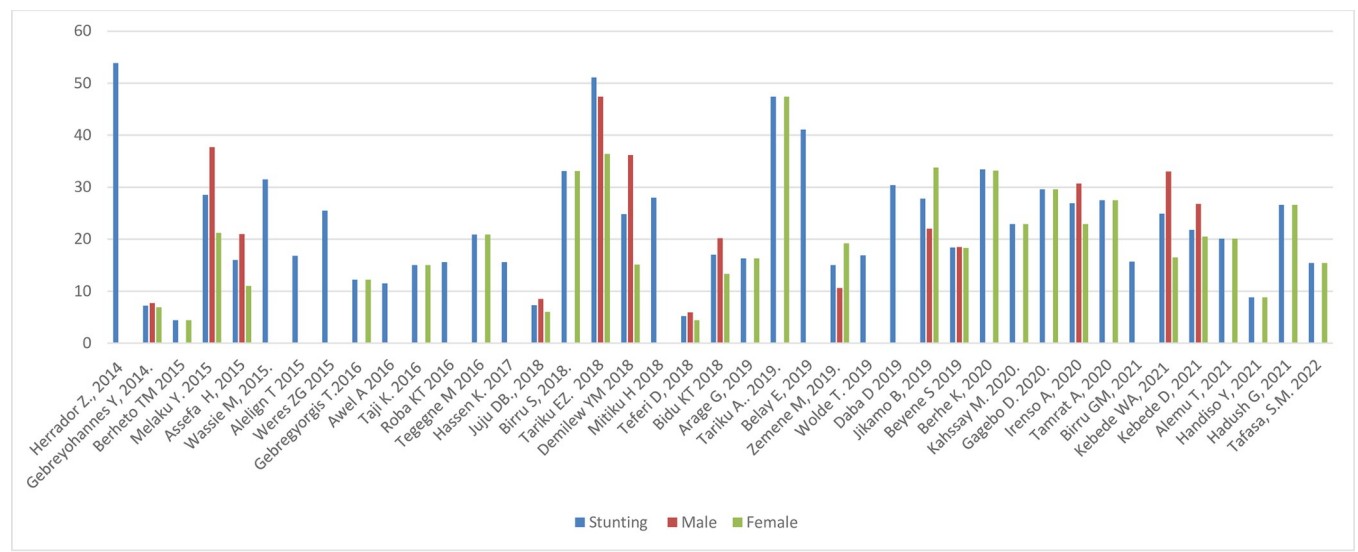

**Fig 6. Trends in adolescent stunting by sex in Ethiopia over the last 7 years.**

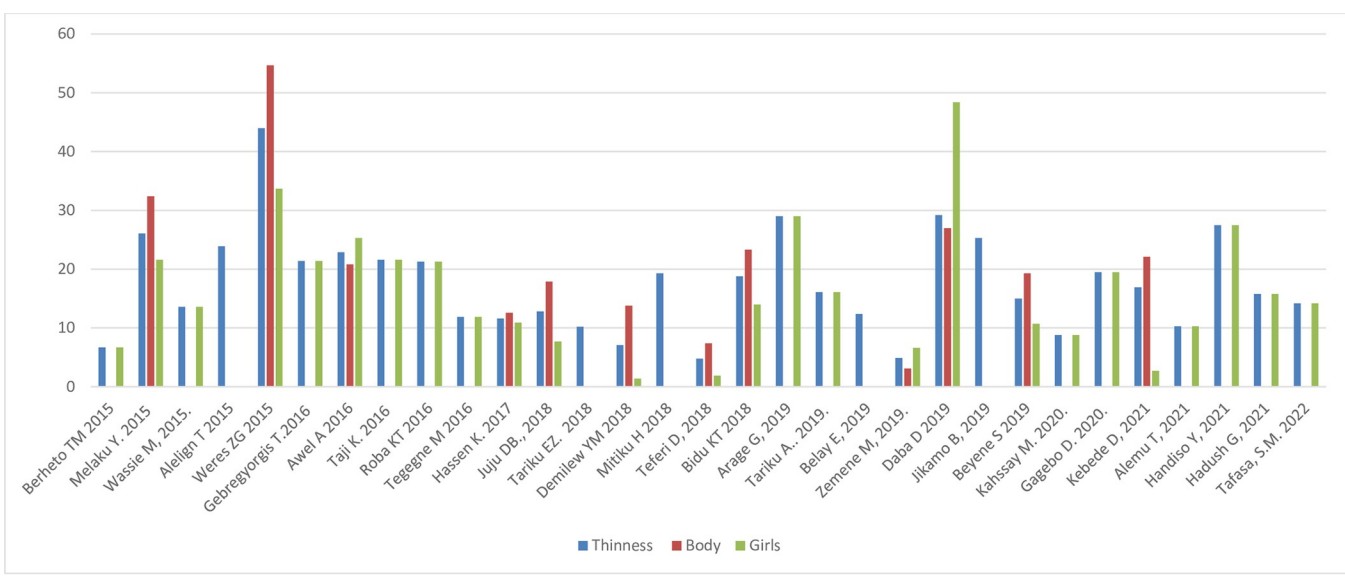

**Fig 7. Trends in adolescent thinness by sex in Ethiopia over the last 6 years.**

CI; 1.3, 3.5) [71]. However, a study from eastern Ethiopia reported a prevalence of anaemia of 32% (HGB<12), 1.8% severe (Hb<7), 3.8% moderate (Hb 7–9.9), and 26.3% mild (Hb 10–11.9) [39].

The national prevalence of anaemia by sex and setting was similar (20.4%) for rural girls and boys while it was 16.7% for girls and 8.6% for boys in urban settings. The national trends in anemia prevalence in girls aged 15–19 years between 2000 and 2016 showed a steady reduction: 23.6% (1.0% severe, 12.7% moderate, and 9.8% mild) in 2005, 13.3% (1.0% severe, 4.7%

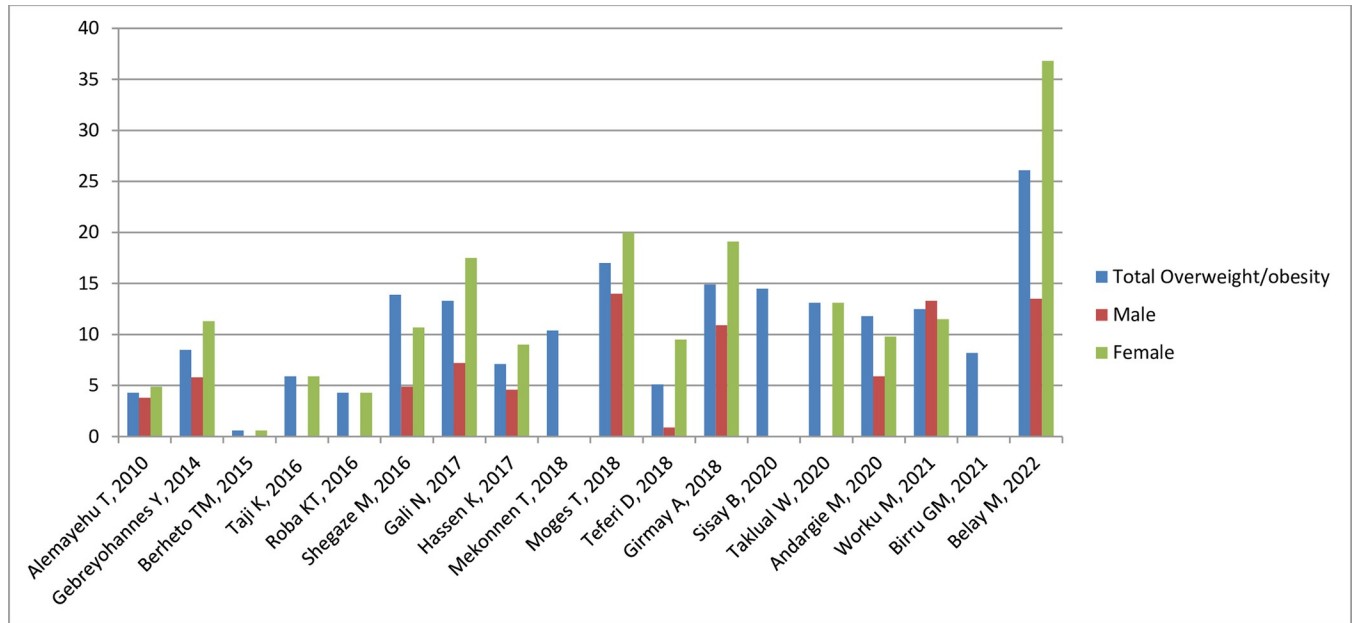

**Fig 8. Trends in adolescent overnutrition by sex in Ethiopia over the last 12 years.**

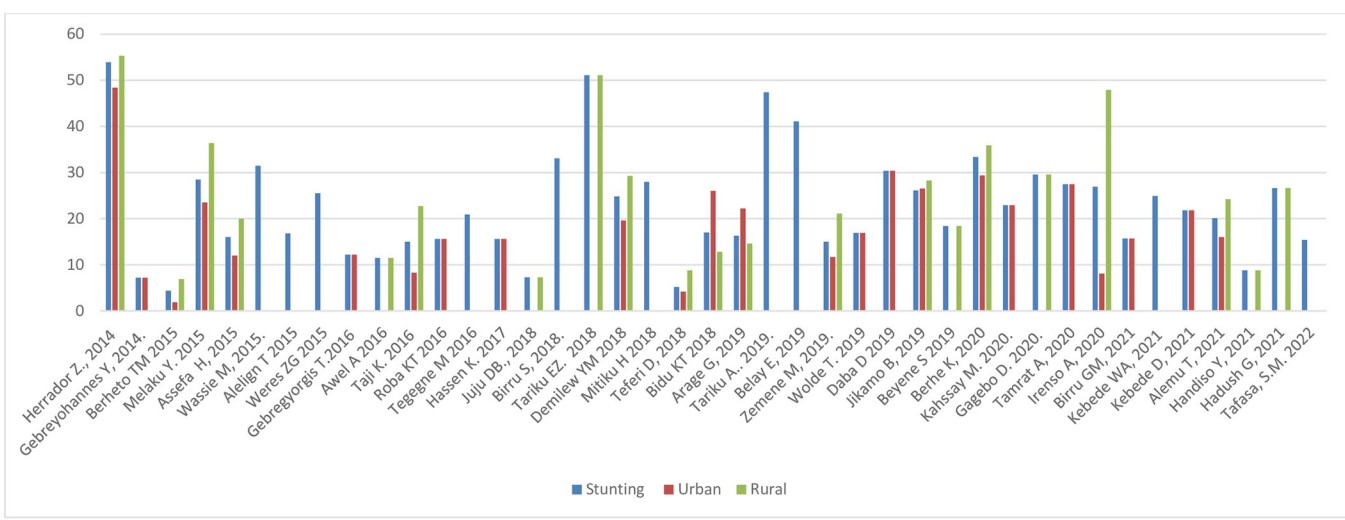

**Fig 9. Trends in adolescent stunting by setting in Ethiopia over the last seven years.**

moderate, and 7.5% mild) in 2011, and 19.6% (0.9% severe, 7.3% moderate and 11.4% mild) in 2016. Equivalent data for boys were 17.7% (0.4% severe, 2.4% moderate and 14.9% mild) in 2011 and 18.2% (severe 0.1%, moderate 3.6%, and mild 14.4%) in 2016 [75].

Two studies in Adama city (Central Ethiopia) by the same authors reported that the average serum vitamin D (25(OH)D) was 54.5nmol/L [73] and the prevalence of vitamin D deficiency (serum 25(OH)D <50 nmol/L) was 42% (female 51.5%, male 29.3%) [70]. Females (AOR1.8; 95% CI: 0.8, 3.8), older adolescents (AOR 1.4; 95% CI: 0.7, 3.1) and urban adolescents (AOR 10.5; 95% CI: 3.9, 28.2) were at higher risk of Vitamin D deficiency [70].

Two papers investigated the prevalence of iodine deficiency; it was 1.88% for severe deficiency (<20μg/L), 25.2% for moderate deficiency (50–99.9 μg/L) and 20.5% for mild deficiency (20–49.9 μg/L), and excess iodine (>300 μg/L) was found in 12.2%, in the age group12-14 years. The prevalence of goitre was 48.9% (girls 65%, boys 35%). Factors include female sex

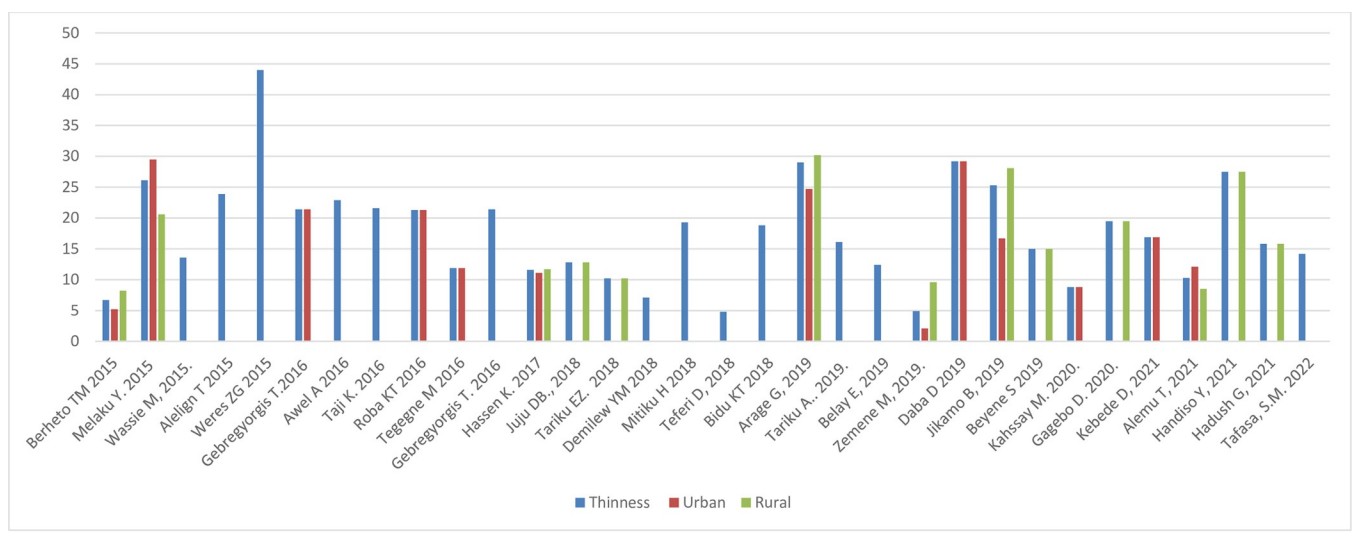

**Fig 10. Trends in adolescent thinness by setting in Ethiopia over the last 6 years.**

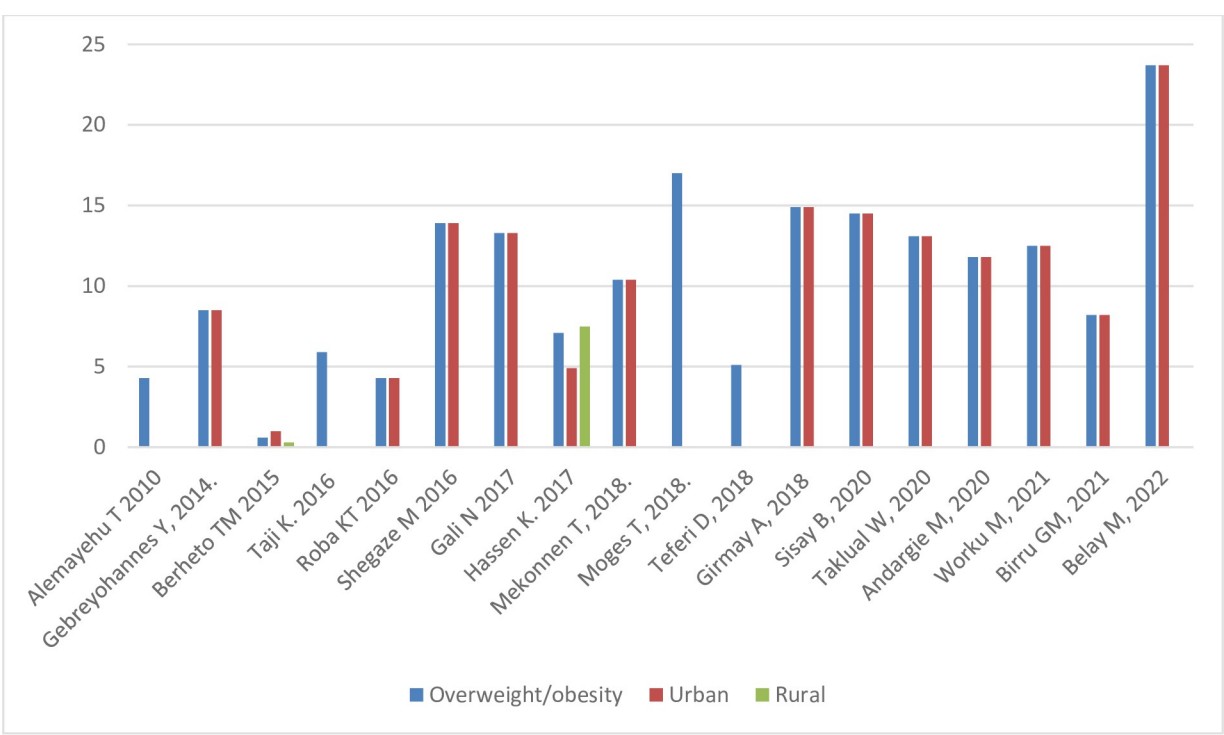

**Fig 11. Trends in adolescent overnutrition by setting in Ethiopia over the last 10 years.**

(AOR = 3.5; 95% CI: 2.6–4.9), living in a temperate climate (AOR = 0.6; 95% CI: 0.4–0.9), and a low frequency of iodized salt use (AOR = 0.5; 95% CI: 0.3–0.7) (67). The magnitude of serum folate deficiency (<6.8nmol/L) was 14.7% [69] (Table 2).

**Diet diversity.** Nine studies [28, 44, 72, 80–85] on a sample of 6 112 adolescents collected dietary data through food frequency questionnaires or 24 hour recalls. Three of these included both girls and boys [28, 44, 81], and the rest included only girls. Six included adolescents from both urban and rural [28, 80–83, 85] settings, two were from urban settings [44, 84] and one was from a rural setting [72]. The lowest prevalence of adequate dietary diversity ((DDS ≥5) was 4.3%, reported in girls from rural areas of Arisi zone (Southeast Ethiopia) in Oromia region [72]. Moreover, a high prevalence of low DDS was reported in different settings: 85.3% in rural and 58.5% in urban adolescents in northwestern Ethiopia [81]. Only 17.5% of adolescents consumed animal source foods in this study [81]. Another study from urban and rural areas of northwestern Ethiopia reported a similar prevalence of low DDS (85.5%) with food secure adolescents more likely to have an adequate DDS (AOR = 1.5, 95% CI 1.03, 2.1) compared to their food-insecure counterparts [82]. In urban settings of northwestern Ethiopia,75.4% (95% CI: 72.3, 78.6) of adolescents had an adequate DDS, higher among those from private schools (AOR = 3.2; 95% CI:1.9,5.3) and from merchant (well-off) families (AOR = 2.4; 95% CI: 1.1,5.5) [84].

Three studies reported mean DDS scores below the average recommended value; 3.3 in southwest Ethiopia [80], 3.5 in northern Ethiopia [83], and 4.3 in Jimma Town [85]. Only one study from an urban setting in south western Ethiopia (Jimma town) reported mean DDS above average (6.97) and cereal based (99.6%) and vegetables (73.9%) diet were the two commonly consumed food types. However, this study did not report the prevalence of low/high DDS [44].

**Table 1. Studies reporting anthropometric measures.**

| First author, y | Main objective (s) | Study design | Setting: Rural/urban | Sample size | Age (y) | Sex (M/F) | Exposure (s) | Outcome (s) | Main findings |
|---|---|---|---|---|---|---|---|---|---|
| Alemayehu T, 2010 [29] | To assess the magnitude of adolescents' undernutrition and its determinants in public schools | Cross-sectional study | Urban | 425 in-school adolescents | 10–19 | M/F | Age, sex, food intake, family livestock ownership | Under-weight and over-weight | Underweight 27.5% (boys 29.8%, girls 24.6%), young adolescents 38.1%, older adolescents 18.6%) Overweight 4.3% (boys 3.8%, girls 4.9%) Underweight predictors: younger age between 10–14 years (AOR = 1.99, 955% CI: 1.01–3.57), household who produce inadequate food supply as a result obliged to purchase (AOR = 2.4, 95% CI: 1.24–4.74) and family possessed no cattle (AOR = 2.4, 95% CI: 1.24–4.74)P<0.05) |
| Gebreyohannes Y, 2014 [31] | To assess and compare nutritional status of adolescents and analyze the risk factors associated with overweight/ obesity in government and private secondary schools | Comparative cross sectional study | Urban | 1024 | 13–19 | M/F | School type | Stunting, underweight, overweight/ obese | Stunting 7.2% (boys 7.7%, girls 6.9%, public school 10.0%, private 4.5%). Underweight 6.2% (boys 9.8%, girls 2.6%, public school 7.0%, private 5.5%), Overweight/obese 8.5% (boys 5.8%, girls 11.3%, public school 4.3%, private 12.7%). Adolescent in private schools are more overweight/obese (AOR 2.2; 95% CI: 1.2–4.2) |
| Herrador Z, 2014 [30] | To determine prevalence of stunting and thinness and their related factors in Libo Kemkem and Fogera, and compare urban and rural areas. Northwest Ethiopia | Cross-sectional Study | Urban/rural | 886 children (259 aged 10–15) | 11–15 | M/F | Residence/setting | Stunting and thinness | Stunting 53.9% (rural 55.3% and urban 48.4%, P-value <0.05) |
| Assefa H, 2015 [22] | To identify socio-demographic factors associated with underweight and stunting among adolescents | 5-year longitudinal study | Urban/rural | 2084 | Mean age 14.8 (SD 1.3) | M/F | Socio-demographic factors | Stunting and underweight | Stunted 16% (boys 21%, girls 11%, urban 12%, semi-urban 16%, rural 20%). Underweight 80.8% (boys 73%, girls 89%%, urban 83%, Semi urban 84%, rural 75%, p-value <0.05), Underweight predictors: male sex (β = -0.7; 95% CI: -0.8, -0.6), age in years (β = 0.1; 95% CI: 0.02, 0.1), attending public school (β = 0.8; 95% CI: 0.02, 1.6) Stunting predictors: male sex (β = -0.2; 95% CI: -0.3, -0.1), attending private school (β = — 1.2; 95% CI: -1.9, -0.5), household income (β = 0.001, 95% CI: 0.001, 0.002), household size (β = -0.02, 95% CI: -0.04, -0.01) |
| Roba A, 2015 [90] | To assess nutritional status and dietary intake of rural adolescent girls and determine pulse and food intake patterns associated with poor nutritional status | Cross-sectional study | Rural | 188 | 15–19 | F | pulse and food intake patterns | Stunting and Underweight | Stunting was 30.9% and underweight was 13.3%. Stunting and underweight associated with low food and nutrient intake. |

*(Continued)*

**Table 1.** (Continued)

| First author, y | Main objective (s) | Study design | Setting: Rural/urban | Sample size | Age (y) | Sex (M/F) | Exposure (s) | Outcome (s) | Main findings |
|---|---|---|---|---|---|---|---|---|---|
| Melaku Y, 2015 [33] | To determine prevalence and factors associated with stunting and thinness | Cross sectional study | Rural/urban | 348 School adolescents | 10–19 | M/F | Sex, sex & setting | Stunting and thinness | Stunting 28.5% (boys 37.7%; girls 21.2%, urban 23.5%, rural 36.4%) Thinness 26.1% (boys 32.4, girls 21.6%, urban 29.5%, rural 20.6%) Mean height-for-age and BMI-for-age Z-scores: -1.49 & -1.29, respectively. Stunting predictors: age 13–15 years (AOR = 2.23; 95% CI: 1.22, 4.08), being male (AOR = 2.53; 95% CI: 1.52, 4.21) and rural residence (AOR = 2.15; 95% CI: 1.20, 3.86). Thinness predictor: male sex (AOR = 1.97; 95% CI: 1.19, 3.25), age 16–19 years (AOR = 0.5; 95% CI: 0.2, 0.9) compared to age 10–12 years |
| Berheto TM, 2015 [32] | To determine urban-rural disparities in the nutritional status of school adolescent girls in the Mizan district, south-western Ethiopia | Comparative cross-sectional study | Urban/rural | 622 (rural 311 and urban 311) | 11–19 | F | Setting | Stunting | Stunting 4.4% (urban 1.9% and rural 6.9%) Thinness 6.7% (urban 5.2% and rural 8.2%) Overweight 0.6% (urban 1% and rural 0.3%) Mean height-for-age Z-score and BMI-for-age Z-score: –0.6 ± (0.9) and –0.4 (1.0) in urban and –0.8 (0.8) and –0.5 (0.9) in rural areas, respectively |
| Weres ZG 2015 [36] | To assess the prevalence of adolescent under nutrition and its associated factors | Cross sectional study | Unstated | 411 | 10–19 | M/F | Age, sex, | Stunting, thinness, underweight | Stunting, 25.5% (Boys 29.6%, girls 21.6%), thinness 44% (boys 54.7%, girls 33.7%) and underweight 55% (boys 65.5%, girls 44.7%) Thinness predictors: younger (10–14 years) age AOR = 4.7; 95% CI = 1.8, 12.1), male sex (AOR = 5.3; 95% CI: 1.7, 16.3) |
| Wassie M, 2015 [34] | To assesses level of low BMI-for- age and height-for- age and their associated factors | Cross-sectional study | Unstated | 1320 | 10–19 | F | Age, dietary diversity, access for nutrition information, and community based nutrition service, food insecurity | Stunting, thinness | Stunting 31.5%, Thinness 13.6% Thinness predictors: age group 10–14 years (AOR = 5.8, 95% CI: 3.3, 10.4), age group 15–17 years (AOR = 2.1, 95% CI: 1.1, 3.9), with poor dietary diversity score (AOR = 2.5, 95% CI: 1.6, 3.8), utilizing community based nutrition service (AOR = 0.7, 95% CI: 0.5, 0.9) Stunting predictors: age group 10–14 years (AOR = 6.1, 95% CI: 4.0,9.2), age group 15–17 (AOR = 1.4, 95% CI: 1.9,2.1), had nutrition and health information(AOR = 1.9, 95% CI: 1.5, 2.6), living in food secured households (AOR:0.7,95% CI: 0.5, 0.8) |
| Alelign T 2015 [35] | To assess the prevalence and factors associated with undernutrition | Cross sectional study | Urban/rural | 403 (209 age 10–14) | 10–14 | M/F | - | Stunting | Stunting 16.8%, underweight 23.9% |

(Continued)

**Table 1.** (*Continued*)

| First author, y | Main objective (s) | Study design | Setting: Rural/urban | Sample size | Age (y) | Sex (M/F) | Exposure (s) | Outcome (s) | Main findings |
|---|---|---|---|---|---|---|---|---|---|
| Awel A, 2016 [38] | To assess nutritional status and associated factors | Cross sectional | Rural | 655 | 10–18 | M/F | Age, sex, family occupation, family size, parental education, daily food intake frequency | Stunting, thinness | Stunting 11.5% (boys 8.4%, girls 14.9%) Thinness 22.9% (boys 20.8%, girls 25.3%) Stunting predictors: female sex (AOR1 = 2.4, 95% CI: (1.3, 4.3); Age 15–18 (AOR = 10.9, 95% CI: 4.8, 24.4), Family size >5 (AOR = 1.9, 95% CI: 1.1, 3.6), lower family wealth index (AOR = 3.2, 95% CI: 1.5, 6.9), Food insecure adolescent (AOR = 2.6, 95% CI: 1.4, 4.9), agro pastoral family occupation (AOR = 2.5, 95% CI: 1.4, 4.7). Thinness predictors: family size >5 (AOR = 1.7, 95% CI: 1.1,2.6), lower family wealth index (AOR = 1.9 (AOR = 1.1, 95%CI:1.1, 3.2), food insecure adolescent (AOR = 2.0, 95%CI: 1.2,3.3) |
| DHS report: Adolescent nutrition, 2000–2016 [75] | To assess the nutritional status of adolescent | Cross-sectional survey | Urban/rural | - | 15–19 | M/F | | Thinness | Thinness: girls = Urban 2.2%, rural 6.8%; Boys = Urban 22.9%, rural 29.6% Overweight: girls = Urban 11.4%, rural (not indicated)? BMI-for-age: Girls thin: 2000 (12.3%), 2005(9.4%), 2011 (8.7%), 2016 (5.7%) BMI-for-age: thinness boys: 2003 (36.6%), 2008 (28.3%) BMI-for-age: Girls overweight: 2000 (2.1%), 2005 (3.9%), 2011 (3.2%), 2016 (4.9%). BMI-for-age: boys overweight: 2003 (0.5)%, 2008 0.8%) Percentage of short stature girls: Urban (10.0%), rural (13.0%) BMI-for-age: Girls; 2000 (20.4%), 2005 (16.6%), 2011 (17.7%), 2016 (12.4%) |
| Roba KT, 2016 [40] | To identify the level of malnutrition and associated Factors | Cross sectional study | urban | 726 | 15–19 | F | Parental education, father occupation, DDS, | Stunting and thinness | Stunting 15.6%, Thinness 21.3%, Overweight 3.3%, obese 1.0%, Thinness predictors: Adolescent from illiterate mother (AOR = 5.4; 95% CI: 4.71–9.1, mothers primary level education (AOR = 1.7; 95% CI:0.9–3.2), FATHERS Illiterate (AOR = 3.1; 95% CI:1.7–5.6), Father primary level education (AOR = 2.4; 95% CI:1.4–4.0), Father ocuupation as daily laborer (AOR = 2.7;95% CI:1.5–4.8), adolescent low DDS (AOR = 2.1; 95% CI:1.5–3.9), |

(*Continued*)

**Table 1.** (Continued)

| First author, y | Main objective (s) | Study design | Setting: Rural/urban | Sample size | Age (y) | Sex (M/F) | Exposure (s) | Outcome (s) | Main findings |
|---|---|---|---|---|---|---|---|---|---|
| Tegegne M, 2016 [42] | To assess the nutritional status and associated factors | Cross sectional study | Urban/rural | 598 | 10–19 | F | Age, setting, parental education, parental occupation, family size, DDS | Stunting, thinness | Stunting 20.9%, thinness 11.9% Stunting predictors: mothers illiterate (AOR = 13; 95%CI: (2.7–18.08), low DDS (AOR = 2.7; 95% CI: 1.5–5.04) Thinness predictors: age≤14 (AOR = 1.7; 95%CI: 1.5–2.6), mother illiterate (AOR = 9.6; 95% CI: 2.6–23.3), mother only read/write (AOR = 7.6; 95% CI: 2.2–19.1), mother primary level education (AOR = 5.2; 95% CI:1.4–17.4) |
| Taji K, 2016 [39] | To assess the nutritional status of adolescent girls | Cross sectional study | Urban/rural | 547 | 10–19 | F | Setting, water source, parental education, parental occupation, | Stunting, thinness, overweight, obesity | Stunting 15% (95% CI: 12.1, 18.3) (urban 8.3%, rural 22.7%), Thin 21.6%, Overweight 4.8% (95% CI: 3.1, 6.)9 Obese 1.1% obese (95% CI: 0.4–2.3) Stunting predictors: fathers with farming occupation (AOR = 2.4; 95% CI: 1.2–4.8), rural residence (AOR = 0.4; 95% CI: 0.2–0.8), younger adolescent (AOR = 0.5; 95% CI: 0.3–0.9) |
| Shegaze M, 2016 [43] | To determine the prevalence of overweight/obesity and associated factors | Cross sectional study | Urban | 456 | 13–19 | M/F | Se, age, family wealth status, physical activity, nutrition knowledge | Overweight/obesity | Overweight 9.7% (95% CI: 6.9, 12.4%), Obesity 4.2% 95% CI: 2.3, 6.0%), Overweight/obesity 13.9% (95% CI: 10.6, 17.1%, boys 4.9%, girls 27.6%) Overweight/obesity predictors: female sex (AOR = 7.3; 95%CI: 3.8, 14.1), private school (AOR = 3.5; 95%CI: 2.0, 6.2), high family wealth (AOR = 4.8; 95%CI: 2.4, 9.8), day time sitting >3 hours (AOR = 6.1; 95%CI: 3.5, 10.8), family size>4 (AOR = 0.3; 95%CI: 0.2, 0.6), low total physical activity level (AOR = 8; 95%CI: 3.9, 16.2), ate sweet food in last 7 days (AOR = 6.3; 95%CI: 3.6, 10.9), meal >3times/day (AOR = 3.0; 1.4, 6.6), better nutrtion knowledge (AOR = 0.2; 95%CI: 0.1, 0.4), |
| Gebregyorgis T. 2016 [91] | To assess the prevalence of thinness, stunting, and associated factors | Cross sectional | Urban /rural | 814 | 10–19 | F | Age, mother education, eating frequency, poor water source, family size, father occupation, father education, wealth index, | Stunting and thinness | Stunting 12.2%, Thinness 21.4% Stunting predictor: Family size >5 [AOR = 2.05 (1.31, 3.23)] and unimproved source of drinking water [AOR = 3.82 (2.20, 6.62)] Thinness predictors: Age of adolescent [AOR = 2.15 (1.14, 4.03)], mother's educational status [AOR = 2.34 (1.14, 4.80)], eating less than 3 meals per day [AOR = 1.66 (1.12, 2.46)], having family size >5 [AOR = 2.53 (1.66, 3.86)] |

(*Continued*)

**Table 1.** (Continued)

| First author, y | Main objective (s) | Study design | Setting: Rural/urban | Sample size | Age (y) | Sex (M/F) | Exposure (s) | Outcome (s) | Main findings |
|---|---|---|---|---|---|---|---|---|---|
| Gali N, 2017 [44] | To determine the prevalence and predictors of obesity and overweight among school adolescents in Jimma town | A school-based cross-sectional study | Urban | 546 | Mean age 15.37 (SD 1.88) | M/F | Age, sex, parental education, dietary intake, school type, family wealth, physical activity, | Overweight/Obesity | Overweight/obesity 13.3% (boys 7.2% and girls 17.5%). Overweight/Obesity predictors: female sex (AOR = 3.4; 95% CI:1.3–9.9]), attending private schools (AOR = 7.5; 95% CI: 2.5–22.3), adolescents from wealthy households (AOR = 3; 95% CI:1.1–8.3]) and. those who were physically inactive (AOR = 3.7; 95% CI:1.1–13.02]) and adolescent with sedentary lifestyles (AOR = 3.6; 95% CI:1.4–9.5) were found to be more obese than their counter peers. |
| Hassen K, 2017 [45] | To investigated the nutritional outcomes of adolescents and their determinants in coffee farming households | Cross-sectional study | Urban/rural | 550 | 10–19 | M/F | Age, residency, family wealth, age dependent family size, parental education, household food insecurity, family size, | Stunting, thinness, overweight/obesity | Stunting 15.6% (girls 16.0%, boys 15.1%, urban 19.8%, rural14.9%), Thinness 11.6% (girls 10.9%, boys 12.6%, urban 11.1%, rural 11.7%), Overweight/obesity 7.1% (girls 9.0%, boys 4.6%, urban 4.9%, rural 7.5%) Stunting predictors: lower teritial of wealth index (AOR = 5.6, 95% CI: 2.6–12.0), Overweight/obesity predictors: middle teritial of wealth index (AOR = 2.7; 95% CI 1.1–6.9) compared to highest wealth index teritial, adolescents in low age dependent family size of 1–2 person/household (AOR = 2.6; 95% CI:1.1–6.2), male sex (AOR = 2.4; 95% CI:1.1–5.1) Thinness predictors: lower wealth teritial (AOR = 5.9; 95% CI: 2.8–12.9), higher family size (AOR = 1.3; 95% CI:1.1–1.5) |
| Bidu KT 2018 [55] | To assess the prevalence and associated factors of undernutrition | Cross sectional study | urban/rural | 640 | 10–19 | M/F | - | Stunting, thinness, | Stunting 17.0% (95% CI: 14%, 20%, boys 20.2%, girls 13.7%, urban 26.0%, rural 12.8%) Thinness 18.8%(95% CI: 15.6%, 21.9%, boys 23.3%, girls 14%) |
| Birru SM, 2018 [47] | To assess prevalence of stunting and associated factors among school adolescent girls in Gondar City | Cross-sectional study | Urban | 812 | 10–19 | F | Age, type od school, parental education, parental occupation, dietary diversity, family wealth index and media exposure | Stunting | Stunting 33.1% (private school 12.1%, public school 38.8%) Stunting predictors: younger (AOR = 0.2; 95% CI: 0.0,0.2), middle age adolescent (AOR = 0.2; 95% CI: 0.2, 0.3), and unsatisfactory media exposure (AOR = 1.7; 95% CI: 1.1, 2.8) and poor mother's education (AOR = 2.8; 95% CI: 1.1, 7.9) |

(*Continued*)

**Table 1.** (Continued)

| First author, y | Main objective (s) | Study design | Setting: Rural/urban | Sample size | Age (y) | Sex (M/F) | Exposure (s) | Outcome (s) | Main findings |
|---|---|---|---|---|---|---|---|---|---|
| Juju D, 2018 [46] | To assess prevalence and factors associated with nutritional status of adolescents in the selected khat and coffee-growing areas | Cross-sectional study | Rural | 234 | 12–18 | M/F | Health problems in the past 30 days | Food insecurity experiences | Stunting 7.3% (boys 8.5%, girls 6.0%). Thinness 12.8% (boys 17.9%, girls 7.7%). Stunting predictors: age 12–14 years (AOR = 3.6; 95% CI, 1.1, 11.5), adolescent from illiterate mothers (AOR = 5.6; 95% CI, 1.6, 20.4). Thinness predictors: Female sex (AOR = 0.4; 95% CI, 0.2, 0.9), dietary frequency <3 times a day (A OR = 4.164; 95% CI, 1.6, 10.7) |
| Teferi D, 2018 [41] | To assess the prevalence of malnutrition and associated factors | Cross sectional study | Urban/rural dominated by urban | 655 | 10–19 | M/F | Age, sex, maternal education, DDS, school type | Stunting, thinness, overweight | Mean height 162.43 cm and weight51.96 kg. Mean HAZ −0.49, and BAZ −0.58 Stunting 5.2% (95% CI: 3.4%,7%, boys 5.9%, girls 4.4%, urban 4.2%, rural 8.8%), thinness 4.8% (95% CI: 3%,6.7%, boys 7.4%, girls 1.9%), and overweight/obesity 5.1% (boys 0.9%, girls 9.5%) Stunting predictors: Maternal secondary educational level (AOR = 0.2; 95% CI: 0.1, 0.9) Thinness predictors: Being male (AOR = 4.1; 95% CI: 2.4,7.0), adolescent from public school (AOR = 0.4; 95% CI: 0.2,0.7), mothers with no formal education (AOR = 4.0; 95% CI: 1.8,8.9), skipping meals (AOR = 1.7; 95% CI: 1.1, 2.7), and illness in 2 weeks prior to survey (AOR = 2.7; 95% CI: 1.5, 4.8) Overweight/obesity predictor: being male (AOR = 0.1; 95% CI: 0.03, 0.2) |
| Zenebe M, 2018 [92] | To examine the effects of school feeding program on dietary diversity, nutritional status and class attendance of school children | Comparative cross-sectional study | Urban/rural | 292 | 10–14 | M/F | School food program | HAZ, BAZ, DDS | Mean (±SD) HAZ score in adolescents with school feeding program was (− 1.45 ± 1.38) compared to those without school feeding program (− 2.17 ± 1.15 which was statistically significant (P < 0.001) adjusted for age, sex, family wealth and parental educational status. |
| Tariku E, 2018 [49] | To assess the prevalence of stunting and thinness and their associated factors among school age children | cross-sectional study | Rural | 389 (137 aged 12–14) | 12–14 | M/F | - | stunting and thinness | Stunting 51.1% (boys 47.4, girls 36.4%), Thinness 10.2%. |
| Mekonnen T, 2018 [48] | To assess the prevalence of overweight/obesity and associated factors | cross-sectional study | Urban | 634 (327 aged 10–14) | 10–12 | M/F | - | Overweight/obesity | Overweight/obese 10.4% |

(*Continued*)

**Table 1.** (Continued)

| First author, y | Main objective (s) | Study design | Setting: Rural/urban | Sample size | Age (y) | Sex (M/F) | Exposure (s) | Outcome (s) | Main findings |
|---|---|---|---|---|---|---|---|---|---|
| Moges T, 2018 [50] | To determine and compare the levels of overweight/obesity among adolescents in private schools with and without adequate play area | Cross-sectional study | Urban | 1,276 | 10–19 | M/F | School play area | Obesity | Overweight/obesity 17.0% (boys 14%, girls 20%, age 10–14 years 16.8%, age 15-19years 17.3%). Mean ± SD BAZ was −0.2± 1.3 Overweight predictor: School with no adequate play area (AOR = 1.6; 95% CI: 1.1, 2.5) |
| Mitiku H, 2018 [52] | To assess the nutritional status of adolescent | Cross sectional study | urban/rural | 1523 (767 aged 10–18) | 10–18 | M/F | - | Stunting and thinness | Stunting 28.0% (in age 10–14 = 26.0%, age 15–18 = 35.3%) Thinness 19.3%(in age 10–14 = 17.6%%, age 15–18 = 39.7%) |
| Girmay A, 2018 [53] | To assess the prevalence of overweight, obesity and associated factors | Cross sectional study | Urban | 950 | 12–15 | M/F | Age, sex, family size, family income, dietary intake, | Overweight/obesity | Overweight/obesity 14.9% (boys 10.9%, 19.1%) positive predictors are female sex (AOR = 1.8; 95% CI:1.2, 2.6)) and taking soft drinks four or more times per week (AOR = 1.0;95%CI: 0.4, 4.6) and lower (<4) family size (AOR = 3.0;95%CI;1.9, 5.0) |
| Demilew Y, 2018 [51] | To assess the prevalence of under nutrition and its associated factors | cross-sectional study | Urban /rural | 424 school adolescents | Mean 16.7 (SD 0.9) | M/F | Sex, parental residence, frequency of dietary intake, water source, family size, illness episode | Under nutrition (stunting and thinness) | Stunting 24.8% (boys 36.2%, girls 15.1%, urban 19.6%, rural 29.3%) Thinness 7.1% (boys 13.8%, girls 1.4%) Stunting predictors: Male sex (AOR = 3.2; 95% CI: (1.7, 5.8), low dietary frequency (1–2 times per day) (AOR = 4.6; 95% CI: 2.6, 8.0), lack of latrine (AOR = 2.7, 95% CI: 1.2, 6.0), and poor hand washing practice (AOR = 3.9; 95% CI: 1.9, 8.1). Thinness predictors: being male [AOR = 11.5; 95% CI: 3.3, 39.5), illness in the last two weeks (AOR = 2.9; 95% CI: 1.2, 7.0), and having more than five family members (AOR = 3.6; 95% CI: 1.3, 9.4) |
| Tariku EZ 2018 | to assess the prevalence of stunting and thinness and their associated factors | Cross-sectional | Rural | 389 (age 12–14, n = 137) | 12–14 | M/F | Sex, age, family size, family income, food security, DDS, parental education, | Stunting and thinness | Stunting 51.1% (boys 47.4%, girls 36.4%) Thinness 10.2% |
| Arage G, 2019 [56] | To determine the prevalence and factor associated with nutritional status of school adolescent girls in Lay Guyint Woreda, Northwest Ethiopia | Cross-sectional study | Urban /rural | 362 | 10–19 | F | Age, residence, mother's occupation, dietary diversity, frequency of dietary intake | Stunting and thinness | Stunting 16.3% (urban 22.2%, rural 14.6%). Thinness 29% (urban 24.7%, rural 30.2%). Stunting predictors: aged 14–15years (AOR = 3.7; 95% CI: 1.9, 7.1), residence in rural areas (AOR = 1.3; 95% CI: 1.2, 2.3), those who did not have snack (AOR = 11.4; 95% CI: 1.5, 17.8) and farming mother's occupation (AOR = 0.1; 95% CI: 0.2, 0.9). Thinness predictors: rural resident (AOR = 2.4; 95% CI: 1.1, 5.1) and adolescents aged 14–15years (AOR = 6.1; 95% CI: 2.2, 17.1). |

(Continued)

**Table 1.** (Continued)

| First author, y | Main objective (s) | Study design | Setting: Rural/urban | Sample size | Age (y) | Sex (M/F) | Exposure (s) | Outcome (s) | Main findings |
|---|---|---|---|---|---|---|---|---|---|
| Belay E, 2019 [58] | To find out the prevalence and determinants of pre-adolescent (5–14 years) acute and chronic undernutrition | Cross sectional study | Urban/rural | 848 (338 aged 10–14) | 10–14 | M/F | - | Stunting and thinness | Stunting 41.1%, thinness 12.4% |
| Beyene S 2019 [63] | To assess the prevalence of undernutrition and associated factors | Cross sectional study | Rural | 1437 | 10–19 | M/F | - | Stunting | Stunting 18.4% (boys 18.5%, girls 18.3%) and thinness 15.0% (boys 19.3%, girls 10.7%) |
| Daba D, 2019 [61] | To assess the prevalence of undernutrition and its associated factors | Cross sectional study | Urban | 312 | 12–18 | M/F | Age, sex, DDS, food intake frequency, water source, substance use, | Stunting and thinness | Stunting 30.4%, thinness 29.2% (Boys 27.0%, girls 48.4%), Thinness predictor: female sex (AOR: 2.55; 95%CI: 1.16–5.63), Ever skipped one or more daily meal per day (AOR: 6.56; 95% CI: 2.25–19.15), low dietary diversity score (AOR: 1.86; 95% CI: 1.05–3.27) and using unprotected water source (AOR: 1.78;95%CI: 1.03–3.05) Stunting predictors; age group 15–18 (AOR: 5.78; 95%CI: 3.20−10.40) and ever used substance (AOR: 3.01; 95%CI: 1.17–7.77). |
| Jikamo B, 2019 [62] | To assess the association between dietary diversity and nutritional status of adolescents | Cross sectional study (Data from the Jimma Longitudinal Family Survey of Youth (JLFSY) | Urban/rural | 2084 | 13–17 | M/F | Age, sex, household food insecurity, adolescent food insecurity DDS, workload | Stunting, thinness | Stunting 27.8% (boys 22%, girls 33.8%, Urban 26.5, Rural 28.3), Thinness 25.3% (urban 16.7%, 28.1%)) Stunning predictors: female sex (AOR = 2.0; 95% CI: 1.6, 2.4), household food insecurity (AOR = 1.7; 95% CI: 0.6, 0.9) Thinness predictor: Household food insecurity (AOR = 1.8; 95% CI: 0.6, 0.8), Rural residents (AOR = 1.6; 95% CI: 1.3, 2.2), Adolescent with higher workload (AOR = 2.6; 95% CI: 1.2, 3.1) |
| Tariku A, 2019. [57] | To assess the prevalence and associated factors of dietary diversity among adolescent girls. | cross-sectional study | Urban/rural | 1550 | 10–19 | F | - | | Stunting 47.4% and thin 16.1% |
| Wolde T, 2019 [60] | To determine the prevalence of stunting and its impact on academic performance | Cross sectional study | Rural/urban | 408 school adolescent | 10–15 | M/F | - | Stunting | Stunting 16.9%. |
| Zemene M, 2019 [59] | To assess the prevalence and its associated factors of nutritional status | Cross-sectional study | Urban/rural | 327 | 10–19 | M/F | Age, sex, residence, family size, water source | | Stunting 15% (boys 10.6%, girls 19.2%, urban 11.7%, rural 21.1%). Thinness 4.9% (boys 3.1%, girls 6.6%, urban 2.1%, rural 9.6%). Stunting predictors: female sex (AOR = 2.2, 95% CI: 1.2, 4.4), rural residence (AOR = 2.5, 95%CI: 1.3, 4.8), and family size of ≥6 (AOR = 3.4, 95% CI:1.7, 7.1) Thinness predictors: Female sex (AOR = 1.8 95% CI: 0.5, 6.5), Rural residence (AOR = 3.7, 95% CI: 1.2, 11.6) |

(Continued)

**Table 1.** (Continued)

| First author, y | Main objective (s) | Study design | Setting: Rural/urban | Sample size | Age (y) | Sex (M/F) | Exposure (s) | Outcome (s) | Main findings |
|---|---|---|---|---|---|---|---|---|---|
| Berhe K, 2020 [64] | To assess the prevalence of undernutrition and associated factors among adolescent girls in Hawzen woreda, Northern Ethiopia | Cross sectional study | Urban/rural | 398 | 10–19 | F | Age, residence, parental occupation, parental education, frequency of dietary intake, family wealth | Stunting, underweight | Stunting 33.4% (urban 29.4%, rural 35.9%), Underweight 32.2% (urban 25.5%, rural 36.3%), Both stunted and underweight 8.8%. Underweight predictors: rural residence (AOR = 1.2; 95% CI: 0.3, 3.1), age 10–13 years (AOR = 0.6; 95% CI: 0.2, 1), unemployed father (AOR = 8.1; 95% CI: 0.5–12.5), unemployed mother (AOR = 2.4; 05% CI: 1.2, 3.6), father illiterate (AOR = 1.4; 95% CI: 1.1, 1.7) Stunting predictors: unemployed father (AOR = 3.2; 95% CI: 1.93–6.4), unemployed mother (AOR = 2.2, 95% CI: 1.1, 3.3), father illiterate (AOR = 1.6; 95% CI: 1.01, 2.2) |
| Gagebo D, 2020 [66] | To assess the prevalence of undernutrition and associated factors among adolescent girls | Cross-sectional study | Rural | 719 | 10–19 | F | Age, family size, parental occupation, parental education, family wealth and dietary frequency | Stunting and thinness | Stunting 29.6% (younger adolescent 25.7%, older 35.6%). Thinness 19.5% (younger adolescent 17.9%, older 21.8%). Stunting predictors: older adolescents (AOR = 2.1; 95% CI: 1.1, 3.9), farmer mother (AOR = 2.4; 95% CI: 1.3, 4.3) and employed mother (AOR = 3.1; 95% CI: 1.4, 6.9)), low household wealth index (AOR = 1.9; 95% CI: 1.3, 2.9), secondary maternal education ((AOR = 0.5; 95% CI: 0.3,0.9), and above secondary maternal education (AOR = 0.3; 95% CI: 0.1, 0.7)). Thinness predictors: father primary education ((AOR = 0.5; 95% CI: 0.3, 0.8) and fathers secondary education (AOR = 0.5; 95% CI: 0.3, 0.8), mother primary education (AOR = 0.6; 95% CI: 0.4, 0.9), adolescent having meal frequency (<2/day) (AOR = 1.9; 95% CI: 1.1, 3.1). |
| Kahssay M, 2020 [65] | To assess the nutritional status of adolescent girls and its associated factors | Cross-sectional study | Urban | 348 | 10–19 | F | Age, family size, dietary diversity, parental occupation, | | Stunted 22.9%, thinness 8.8%. Stunting predictors: adolescent age 14–15 years (AOR = 1.4, 95% CI: 1.1–4.3), and dietary diversity score of <4 food groups (AOR = 2.2, 95% CI: 1.4–4.5). Thinness predictors: dietary diversity score of <4 food groups (AOR = 1.8, 95% CI: 1.1–4.4) and low food consumption (AOR = 3, 95% CI:1.2–7.9) |

**Table 1.** (Continued)

| First author, y | Main objective (s) | Study design | Setting: Rural/urban | Sample size | Age (y) | Sex (M/F) | Exposure (s) | Outcome (s) | Main findings |
|---|---|---|---|---|---|---|---|---|---|
| Taklual W, 2020 [93] | Aimed at assessing nutritional status and associated factors among female adolescents | school-based cross-sectional study | Urban | 682 | 14–19 | F | Age, family size, religion, ethnicity, parental occupation, parental education, family wealth, types of staple diet, diet diversity, menarche onset | Underweight, overweight, and obesity | Underweight 15%, overweight 8.4%, and obesity 4.7% Underweight predictors: Age groups of 14–16.5 years (AOR: 1.7, 95% CI: 1.03–2.69), family size ≥ 4 (AOR: 2.8, 95% CI: 1.05–4.99), participants who did not eat meat once per week (AOR: 1.6, 95% CI: 1.90–2.82), and no onset of menarche (AOR: 4.4, 95% CI: 1.21–15.75) Overweight predictors: family monthly income above 6500 ETB (AOR: 12.7, 95% CI: 2.47–65.62), consumption of meat two or more times per week (AOR: 2.07, 95% CI: 1.47–9.14), and consumption of fruit at least once a week (AOR: 0.20, 95% CI: 0.05–0.78) |
| Irenso A, 2020 [94] | To assess the magnitude and factors associated with adolescent linear growth and stunting | Cross-sectional | Urban/rural | 2010 | 10–19 | M/F | Age, sex, residence, hygiene, | Linear growth and stunting | Overall stunting 26.9% (Boys 30.7, girls 22.9; Urban 8.1%, Rural 47.9%. Significant interaction between residence and sex on the risk of stunting [AOR = 4.17 (95% CI 2.66, 9.9), P < 0.001], and height-for-age z score (HAZ) (b = 0.51, P < 0.001). In urban adolescents, older age (18 to 19 years) was negatively associated with linear growth (b = 0.29; P < 0.001). In rural setting, hand washing practice after toileting was positively associated with HAZ (0.62; P < 0.001) and with lower risk of stunting [AOR = 0.51 (95% CI 0.34, 0.76)]. Urban females had significantly higher HAZ than urban males [b = 0.52; P < 0.01)], and a significantly lower risk of stunting [AOR = 0.29 (95% CI 0.18, 0.48)]. |
| Tamrat A, 2020 [95] | Aimed at determining the prevalence of stunting and its associated factors | school-based cross-sectional study | Urban | 662 | 10–14 | F | Age, religion, grade level, parental education, parental occupation, family size | Stunting | Stunting 27.5%. Stunting predictors: being grade 5 student [AOR; 95% CI: 1.90; 1.13–3.20], less than three meal a day [AOR; 95% CI: 2.37; 1.60–3.50], household food-insecurity [AOR; 95% CI: 2.52; 1.70–3.73]. Stunting preventive factors: Government employed mothers [AOR; 95% CI: 0.48; 0.26–0.89] or merchants [AOR; 95% CI: 0.43; 0.28–0.67] |

(*Continued*)

**Table 1.** (Continued)

| First author, y | Main objective (s) | Study design | Setting: Rural/urban | Sample size | Age (y) | Sex (M/F) | Exposure (s) | Outcome (s) | Main findings |
|---|---|---|---|---|---|---|---|---|---|
| Andargie M, 2020 [96] | to assess the magnitude and associated factors of overweight and obesity among public and private secondary school adolescents in Mekelle city | school-based comparative cross-sectional between private and public school adolescents | Urban | 858 | 14–19 | M/F | Age, type of school, religion, family size, birth order, grade level, physical activity, food frequency, type of transport to school, nutrition knowledge, parental occupation, parental education, parental wealth | Overweight and obesity | Overall overweight and obesity 7.8% (boys 5.9(, girls 9.8%, private school 11.8% and public schools 3.9%) Overweight/obesity predictors: Consuming dinner not daily [AOR = 5.3:95% CI = 1.93–14.6] and working moderate-intensity sports at least 10 minutes/day continuously [AOR = 0.19:95% CI = 0.04–0.9] were associated factors of overweight and obesity in public school adolescent students. Being female [AOR = 2.03:95% CI = 1.08–3.8], time taken from home to public physical activities ≤ 15 minutes [AOR = 3.6:95% CI = 1.13–11.51], using transport from school to home [AOR = 2.2:95% CI = 1.06–4.18] and good knowledgeable adolescents [AOR = 0.5:95% CI = 0.27–0.9] were associated factors of overweight and obesity in private schools. |
| Sisay B, 2020 [89] | To evaluate the performance of MUAC to identify overweight (including obesity) in the late adolescence period | Cross-sectional study | Urban | 851 | 15–19 | M/F | - | | Overweight 11.2% (95% CI; 9.2–13.5%), Obesity 3.3% (95% CI; 2.3–4.7%) BMI Z score 0.44 (±1.2) |
| Worku M, 2021 [97] | To assess the prevalence and associated factors of overweight and obesity | nstitution-based cross-sectional study | Urban | 551 | 10–19 | M/F | Age, sex, school type, DDS, religion, parents occupation, family wealth status | overweight and obesity | Mixed overweight and obesity 12.5% (Boys 13.3%, Girls 11.5%) Overweight/obesity predictors: Having self-employed mothers (AOR: 4.57; 95% CI: 1.06, 19.78), having government-employed mothers (AOR: 6.49; 95% CI: 1.96, 21.54), and having school feeding access (AOR: 0.44; 95% CI: 0.26, 0.76) |
| Kebede D, 2021 [98] | To assess the prevalence and associated factors of stunting and thinness | school-based cross-sectional study | Urban | 397 | 10–19 | M/F | Age, sex, family wealth, grade, place of residence, religion, parental education, parental occupation, family size, DDS | Stunting and thinness among | Stunting 21.8% (Boys 26.8%, girls 20.5%) & thinness 16.9% (Boys 22.1%, girls 2.7%) Stunting predictors: having a family monthly income of less than $28.37 (P = 0.044) and having less than four dietary diversity (P = 0.021) Thinness predictors: Early adolescent age, being male, having a family monthly income of less than $28.37, having a family monthly income between $28.37 and $56.74 (P = 0.021) (35.25 Birr = 1 USD) and using clean water (P = 0.045) |

*(Continued)*

**Table 1.** (Continued)

| First author, y | Main objective (s) | Study design | Setting: Rural/urban | Sample size | Age (y) | Sex (M/F) | Exposure (s) | Outcome (s) | Main findings |
|---|---|---|---|---|---|---|---|---|---|
| Alemu T, 2021 [99] | Aimed at comparing the rural and urban prevalence's ofstunting and thinness and their associated factors | ommunity-based comparative cross-sectional study | Urban/ Rural | 792 | 10–19 | F | Age, educational status, residence, parental occupation, parental education, famlily sixe, family wealth, religion | Stunting and thinness | Stunting 20.1% (Urban 16%, Rural 24.2%), Thinness 10.3% (Urban 12.1%, rural 8.5%) Stunting predictors: Food insecurity [AOR: 1.95 (95% CI: 1.01, 3.78)] Stunting predictors in urban settings: early age adolescent [AOR:3.17 (95% CI:1.445,6.95)] Stunting predicators rural settings: lack of latrine [AOR: 1.95 (95% CI: 1.11, 3.43)], lowest media exposure [AOR: 5.14 (95% CI: 1.16, 22.74)], lower wealth class [AOR:2.58 (95% CI: 1.310, 5.091)], and middle wealth class[AOR: 2.37 (95% CI: 1.230, 4.554)] Thinness predictors Rural settings: Middle age adolescent groups [AOR: 3.67 (95% CI: 1.21, 11.149)]. Thinness predictors Urban setting: early age adolescent [AOR: 8.39 (95% CI: 2.48–28.30)]. |
| Handiso Y, 2021 [100] | To assess the nutritional status and associated factors among adolescent | community-based cross-sectional study | Rural | 843 | 10–19 | F | Age, religion, school grade, family size, family income nutrition edutaion, dewarming, nutrtion service receved, | Thinness, stunting | Stunting 8.8%, Thinness 27.5%, Predictors of thinness: [AOR; 95% CI = 2.91; 2.03–4.173], large family size [AOR; 95% CI = 1.6; 1.11–2. 40], low monthly income [AOR; 95% CI = 2.54; 1.66–3. 87], not taking deworming tablets [AOR; 95% CI = 1.56;1.11–21], low educational status of the father [AOR; 95% CI 2.45; 1.02–5.86], source of food only from market [AOR; 95% CI = 5.14; 2.1–12.8], Predictors of stunting: lack of service from health extension workers [AOR; 95% CI = 1.72; 1.7–2.4], and not washing hand with soap before eating and after using the toilet [AOR; 95% CI = 2.25, 1.079–4.675] |
| Hadush G, 2021 [101] | to assess prevalence of nutritional status and associated factors among adolescent girls | school-based cross-sectional study | Rural | 736 | 10–19 | F | Age, family size, parents occupational status, parents educational status, family wealth status, household food insecurity | Thinness and stunting | Stunting 26.6%, Thinness 15.8%, Stunting predictors: being at an early adolescent age (AOR = 1.96, 95% CI 1.02–3.74), household food insecure (AOR = 2.88, 95% CI 1.15–7.21), menstruation status (AOR = 2.42, 95% CI 1.03–5.71), and availability of home latrine (AOR = 3.26, 95% CI 1.15–4.42). Thinness predictors: early age adolescent (AOR = 2.89, 95% CI 1.23–6.81) |

(*Continued*)

**Table 1.** (Continued)

| First author, y | Main objective (s) | Study design | Setting: Rural/urban | Sample size | Age (y) | Sex (M/F) | Exposure (s) | Outcome (s) | Main findings |
|---|---|---|---|---|---|---|---|---|---|
| Kebede WA, 2021 [102] | aimed at assessing the magnitude of stunting and associated factors among adolescent students | School survey | Urban/rural | 424 | 14–19 | M/F | Age, sex, religion, residence, family economy, parental education, grade level, water and sanitation, | Stunting | Stunting 24.9% (Boys 33%, girls 16.5%) Stunting predictors: male sex [AOR = 2.1; 95% CI: 1.73–5.90], meal frequency (<3/day) [AOR = 4.6; 95% CI: 2.61–8.24], infrequent hand washing practice [AOR = 3.6; 95% CI: 1.30–9.40], absence of latrine facility (AOR = 5.51; 95% CI: 3.03–9.9), and consumption of unsafe water [AOR = 2.8; 95% CI: 1.35–6.19]. |
| Birru GM, 2021 [103] | to assess malnutrition and the associated factors among adolescents | School survey | Urban | 365 | 14–19 | M/F | Age, sex, parental marital status, DDS, food frequency, diet quality, mother occupation, snack intake | Stunting, underweight,/ thinnessoverweight/ obesity | Stunted 15.7%, Underweight 6.3%, and overweight/obesity 8.2%. Stunting predictor: Daily snack intake (AOR = 0.38, 95% CI: 0.20, 0.71), and inadequate diet quality (AOR = 3.36, 95% CI: 1.15, 7.82) underweight/thin: Being a male (AOR = 2.76, 95% CI: 1.03, 7.44) and meal consumption <3 times/day (AOR = 4.21, 95% CI: 1.35, 13.11) Overweight/Obesity: Dietary diversity score<5 (AOR = 0.35, 95% CI: 0.13, 0.89) |
| Kedir S, 2022 [104] | Aimed at identifying context-specific determinants of overweight and/or obesity among adolescents | School-based unmatched case-control study design | Urban | 297 | 10–19 | M/F | Sex, Age, wealth, soft drinks consumption, physically activity, screen time, nutritional knowledge, family size, parental education, diet diversity, fast food consumption | Overweight/ obesity. | High socioeconomic status [AOR = 5.8, 95% CI (2.66, 12.5)], consumed soft drinks 3and more times per week [AOR = 3.7, 95% CI (1.8, 7.3)], physically inactive [AOR = 4.4 95% CI (1.68, 11.6)], spent free time by watching television/ movies for 3and above hours per day [AOR = 8.6, 95% CI (4.3, 17)] and with poor nutritional knowledge [AOR = 3.4, 95%CI (1.7, 6.9)] were significantly associated with overweight/ obesity. |
| Tafasa, S.M. 2022 [105] | to assess the prevalence of undernutrition and its associated factors among school adolescent girls | School based study | Urban/rural | 587 | 10–19 | F | Age, religion, place f residence, family marital status, educational level parental education, family size, physical activity, DDS, food frequency, water source, nutrition knowledge, illness episodes, menstruation | Stunting and thinness | Stunting 15.4%; thinness 14.2%, Stunting predictors: Less than 3 meal/day [AOR = 3.62, 95% C.I (2.16, 6.05)], attending lower grades [AOR = 2.08, 95% C.I (1.07, 4.04)] and did not started menstruation [AOR = 1.71, 95% C.I (1.06, 2.73)] Thinness predictors: vigorous physical activities [AOR = 2.51, 95% C.I (1.14, 5.54)], low dietary diversity score [AOR = 4.05, 95% C.I (1.43, 11.46)] and younger adolescent (10–14 yrs) [AOR = 3.77, 95% C.I (1.06, 13.37)] |

(*Continued*)

**Table 1.** (Continued)

| First author, y | Main objective (s) | Study design | Setting: Rural/ urban | Sample size | Age (y) | Sex (M/ F) | Exposure (s) | Outcome (s) | Main findings |
|---|---|---|---|---|---|---|---|---|---|
| Belay M, 2022 [106] | To determine the magnitude of overnutrition and associated factors among school adolescents in Diredawa city | School based survey | Urban | 498 | 10–19 | M/ F | Age, sex, meal preference, type of school, snack intake, physical activity, parental education, parental occupation, household wealth status | Overnutrition | Overnutrition of 26.1% (boys 13.5%, girls 36.8%); Overweight 23.7% and Obesity 2.4% Overnutrition Predictors: Being female (AOR = 3.32; 95% CI: 1.65–6.63), attending at private school (AOR = 4.97; 95% CI: 1.72–14.35), having sweet food preferences (AOR = 6.26; 95% CI: 3.14–12.5), snacking (AOR = 3.05; 95% CI: 1.11–8.36), sedentary behavior (AOR = 3.20; 95% CI: 1.67–6.09), and eating while watching TV (AOR = 2.95; 95% CI: 1.47–5.95) |

In terms of site preference for nutrition interventions, as reported by adolescents, schools (45%), health centers (27%) and health posts (26%) were the preferred public facilities for provision of iron supplements to school adolescents, while schools (11%), health centers (47%) and health posts (41%) were the preferred public facilities for provision of iron supplements to out-of-school adolescents [83]. In the same study, it was indicated that a lack of nutrition messages specifically for young people, low community awareness about adolescent nutrition, religious and cultural influences, perceiving iron as a contraceptive than a nutrition product, and lack of confidence in the supplementation value of iron tablets are barriers to the uptake of adolescent nutrition interventions in northern Ethiopia (Table 3).

**Food insecurity.** Seven studies [23–27, 46, 86] with a sample size of 10 866 adolescents assessed food insecurity, of which five came from the Jimma Longitudinal Family Survey of Youth (JLFSY) study [23–27] which followed 2 084 adolescents over three years. The remaining two studies were cross sectional surveys with a sample size of 784 adolescents in areas producing Khat (a common evergreen plant in eastern Africa used for its psychoactive properties) and coffee. Food insecurity was assessed using the adolescent food insecurity assessment scale adopted from household food security questionnaire [107], which enquires about their experience or concern about access to food or money.

The prevalence of food insecurity in Jimma zone (urban and rural settings) was 59.6% [86]. In this study, female adolescents (AOR = 2.2, 95%CI:1.4, 3.5), household food insecurity (AOR = 9.4,95%CI:5.5, 16.2), a male head of household (AOR = 2.8, 95% CI:1.4, 5.3), a high dependency ratio (AOR = 2.5, 95% CI: 1.5, 4.5), a household head with no formal education (AOR = 4.9, 95% CI: 2.6,9.2) and a family which does not own farming land (AOR = 2.5, 95% CI: 1.2, 5.0) were positively associated with food insecurity [86]. The prevalence of food insecurity in Khat and Coffee producing areas of Sidama zone was lower at 38.0% (boys 40.2% and girls 35.9%; p-value 0.412) [46]. The prevalence of food insecurity was higher in coffee (43%) compared to khat producing areas (32.4%) [46].

The Jimma Longitudinal Family Survey of Youth (JLFSY) study was started in 2005 and has 3-yearly follow-ups [23–27]. In this longitudinal cohort, 20.4%, 48.4% and 20.6% of

**Table 2. Studies focusing on micronutrients.**

| Authors (y) | Objective (s) | Study design | Settings | Sample | Age | Sex | Exposure (s) | Outcomes (s) | Main findings |
|---|---|---|---|---|---|---|---|---|---|
| Desalegn D, 2014 [74] | To determine the prevalence, severity, and predictors of nutritional IDA | Cross sectional study | Urban | 586 (269 aged 10–12) | 10–12 | M/F | - | Iron deficiency anemia (Hb <12 g/dl) | Iron deficiency anemia 32.7% |
| Wakao T, 2015 [70] | To determined vitamin D deficiency and its predictors | Cross sectional study | Urban/rural | 174 | 11–18 | M/F | Age, sex, parental education, wealth index | vitamin D deficiency (25 (OH)D <50 nmol/L) | Vitamin D deficiency was 42% (girls 51.5%, boys 29.3%). Females (AOR = 1.76; 95% CI: 0.8, 3.8), older adolescent (AOR = 1.4; 95% CI: 0.7, 3.1) and urban residence (AOR = 10.5; 95% CI: 3.9, 28.2) are at higher risk of Vitamin D deficiency. |
| DHS report, 2016 [75] | To assess adolescent nutrition, including anaemia | Repeated cross-sectional survey | Urban/rural | - | 15–19 | M/F | Sex and setting | Anemia ((Hb <12 g/dl) | Anemia: urban girls 16.7%, rural girls 20.4%, Urban boys (8.6%), rural boys (20.4%). Anemia trends: in girls 15–19 years: 2005 (23.6; severe 1.0, moderate 12.7, mild 9.8), 2011 (13.3; severe 1.0, moderate 4.7%, mild 7.5%), 2016 (19.6; severe 0.9%, moderate 7.3%, mild 11.4%). Anemia trend: in boys 15–19 years: 2011: 17.7% (0.4% Severe, 2.4 moderate and 14.9 mild), 2016 18.2% (severe 0.1%, moderate 3.6%, and mild 14.4%). |
| Ministry of Health, 2016 [69] | Ethiopia National Micronutrient Survey to estimate the prevalence of selected micronutrient (Iron, Folate, vitamin A, Retinol, Zinc, Iodine, Vitamin B12) deficiencies | Cross-sectional study | Urban/rural | 722 | 12–19 years | M/F | Age | Anemia, level of vitamin A, Vitamin B12, deficiencies of Zinc, iodine, folate | Anemia ((Hb <12 g/dL):14.9% (moderate (Hb 8–12 g/dL14.4%, severe (Hb<8g/dL 0.5%) in age of 12–14 years and 11.8% (10.5% moderate and 1.3% severe) in age 15–19 years. Iron deficiency (Ferritin<15) was 8.6%. Iron deficiency anemia (Ferritin) was 2.6% and (STFR) was 4.3% in age 12–14 years. Mean (SD) Vitamin A status was 1.20 ±0.35 and % (Retinol <0.7 μmol/l) was 6.3% in age 12–14 years. Iron deficiency (Ferritin<15) was 10.0%, (STFR<4.4) 83.8%, and IDA (Ferritin) 3.2%, IDA (serum transforming receptor) was 4.7%, vitamin A mean value 1.40 ±0.43, and % (Retinol <0.7 μmol/l) was 3.2% in age 15–19 years. Deficiencies: Zinc 38%, Iodine 1.88% for severe deficiency (<20μg/L), 25.2% for moderate deficiency (50–99.9 μg/L) and 20.5% for mild deficiency (20–49.9 μg/L). Excess iodine (>300 μg/L) 12.2% in age group 12–14 years. Deficiencies: Serum folate (<6.8nmol/L) 14.7%, Vitamin B12 (<203 pg/ml) 13.6%, iodine (Severe deficiency(<20 μg/L) 1.9%, moderate deficiency (50–99.9 μg/L) 28.6%, mild deficiency (20–49.9 μg/L) 21.9%, and Excess (>300 μg/L) 7.9% in age group 15–19 years. |
| Teji K, 2016 [39] | To assess the prevalence of anaemia and nutritional status of adolescent girls | Cross-sectional study | Urban/rural | 547 | 10–19 | F | - | Anemia ((Hb <12 g/dl) | Anemia 32% (HGB<12), severe 1.8% (HGB<7), moderate 3.8% (HGB 7–9.9), and mild 26.3% (HGB 10–11.9) |

*(Continued)*

**Table 2.** (Continued)

| Authors (y) | Objective (s) | Study design | Settings | Sample | Age | Sex | Exposure (s) | Outcomes (s) | Main findings |
|---|---|---|---|---|---|---|---|---|---|
| Getaneh Z, 2017 [68] | To assess the prevalence and associated factors of anemia | Cross-sectional | Urban | 523 (332 aged 11–14) | 11–14 | M/F | - | Anemia (Hb <12 g/dl) | Anemia 13.4% |
| Workie S, 2017 [67] | To assess the prevalence of iodine deficiency disorder | Cross-sectional study | Urban/rural | 718 | 10–20+ | M/F | Age, sex, setting, use of iodized salt | Iodine deficiency disorder (as measured by thyroid gland Enlargement) | Goiter 48.9% (boys 35.1%, girls 65.2%). Grade-1 goiter 36.9% and Grade-2 goiter 11.9%. Goiter is associated positively with girls (AOR = 3.5; 95%CI: 2.6–4.9) and negatively with regular use of iodized salt (AOR = 0.5; 95%CI: 0.3–0.7) |
| Gonete K, 2017 [71] | To assessed the prevalence and associated factors of anemia | Cross-sectional study | Urban/rural | 462 | 15–19 | F | Setting, dietary diversity, household food insecurity, source of water | Anemia (Hb <12 g/dl) | Overall anemia was 25.5%, (95% CI: 21.4, 29.2) with mild anemia 92.4%, moderate 5.9% and severe 1.7% Odds of having anemia among those with inadequate DDS was 2.1 higher (AOR = 2.1; 95% CI: 1.3, 3.5). |
| Wakayo T, 2018 [73] | To evaluate the association between Serum Vitamin D levels of 25(OH)D and handgrip strength | Cross-sectional study | Urban/rural | 174 | 11–18 | F | - | Serum Vitamin D level | Average serum 25(OH)D was 54.5 + 15.8 nmol/L. |
| Seyoum Y, 2019 [72] | To determine the prevalence of iron deficiency, low iron stores, and anemia and characterize selected risk factors | Cross-sectional study | Rural | 257 | 15–19 | F | - | Anemia (Hb <11 g/dl) | Anemia 8.7% (Hb <11 g/dL) and clinical iron deficiency 8.7% (Serum Ferretin <15 μg/L), but 41% had marginal iron stores (SF <50 μg/L). |
| Mengistu G 2019 [77] | To assess the prevalence of anemia and associated factors | Cross sectional | Urban | 423 | 10–19 | F | Age, DDS, family zise, family income, | Iron deficiency anemia (Hb <12 g/dl) | Anemia 11.1%. Predictors are: family size>5 [AOR = 3.2, 95%CI: 1.3–7.9), lower average family income (AOR = 10; 95%CI; 2.5–41.3). |
| Demelash S 2019 [78] | To assess the prevalence of anemia and its associated factors | Cross-sectional study | Urban | 594 | 15–19 | F | - | Anemia (Hb<12 g/dl) | Anemia prevalence 21.1% (CI: 17.4, 24). |
| Gebreyesus SH 2019 [79] | To evaluate the prevalence of anaemia | Cross sectional study | Urban/rural | 1323 | 10–19 | F | Age, residency, food insecurity | Anemia (Hb< 12.0 g/dl) | Anemia 28.8% (Urban 31.6%, rural19%) Anemia predictors: Younger (10–14 years) adolescent (AOR = 2.0; 95% CI: 1.1, 3.8) |
| Regasa RT, 2019 [80] | To determine the status of anemia and its anthropometric, dietary and socio demographic determinants | Cross-sectional study | Urban/rural | 448 | 10–19 | F | Age, residency, family size, parental education family wealth status, DDS, | Anemia (Hb< 12.0 g/dl, mild (10–11.9 g\dl), moderate (7.0–9.9 g/dl,) or severe (<7 g\dl) | Anemia 27% (95% CI: 22.9–31%, mild 23% and moderate 4%) Associated factors: age (younger 10–14 years 38.6%, older age 15–19 years 12.6%, p-value <0.05), settings (rural 45.3%, urban 12.5%, p-value <0.05) |
| Gebremichael G, 2020 [76] | To investigate the prevalence of goiter and associated factors | Cross-sectional study | Urban/rural | 576 | 10–19 | M/F | Sex, age, family history of goiter, residence, use of iodized salt, DDS, altitude | Goiter | Goiter 42.5% (95% CI: 38.4, 46.7; boys 34%, girls 50.9%) Goiter predictors: Being female (AOR = 1.8; 95% CI: 1.2, 2.9), family history of goiter (AOR = 3.6; 95% CI: 2.3, 5.7), lack of meat consumption (AOR = 2.5; 95% CI: 1.2, 5.3), lack of milk consumption (AOR = 2.2; 95% CI: 1.2, 4.0), and inadequate use of iodized salt (AOR = 7.1; 95% CI: 3.8, 12.9) |

**Table 3. Studies focusing on dietary diversity score (DDS).**

| Author (y) | Objective (s) | Study design | Settings | Sample | Age | Sex | Exposure | Outcomes | Main finding including description for each article |
|---|---|---|---|---|---|---|---|---|---|
| Herrador Z, 2015 [81] | To identify associated factors for low dietary diversity and lack of consumption of animal source food (ASF) | A cross-sectional survey with an additional follow up observation study | Urban/rural | 886 (320 aged 10–18) | 10–18 | M/F | - | DDS | Low DDS; rural 85.3%, urban 58.5%. Consumption of animal source food 17.5%. |
| Mulugeta A, 2015 [83] | To examine means of reaching adolescent girls for iron supplementation | Cross-sectional study | Urban/rural | 828 | 15–19 | F | - | DDS | Prevalence of low, medium and high DDS was 54%, 42.9% and 3.1%, respectively. Mean DDS: 3.5 Schools (45%), health centers (27%) and health posts (26%) were the preferred public facilities for provision of iron supplements to school adolescents. Schools (11%), health centers (47%) and health posts (41%) were the preferred public facilities for provision of iron supplements to out of school adolescent |
| Tamiru D, 2016 [28] | To assess the effectiveness of school-based health and nutrition Intervention, supported with backyard gardening, on dietary diversity | Quasi experimental study | Urban/rural | 1000 | 10–19 | M/F | School-based health and nutrition intervention supported with backyard gardening | DDS | Prevalence of adequate DDS in the intervention group across time was 34.8% at baseline, 65.6% at midline and 74.7% at end line. Prevalence of adequate DDS in the control group across time was (32.1%) at baseline, (49.4%) at midline and (48.8%) end line. Effect of the intervention between intervention and control group: there was statistically significant difference at mid-point (F = 5.64, p = 0.042) and end (F = 5.85, p b 0.001). |
| Melaku Y 2017 [85] | To assess optimal dietary practices and nutritional knowledge | Cross sectional study | Urban/rural | 455 | 14–19 | F | Age, residency, maternal education, father occupation, family size | DDS | Mean (±SD) DDS was 4.3 ± 1.4. Low (<5) DDS 61.3% (Urban 37.7%, rural 50.0%) Low DDS predictors: attending government school (AOR = 5.2; 95% CI: 2.9,9.4), mothers illiterate (AOR = 7.7; 95%CI:3.4, 17.2), PRIMARY level education (AOR = 5.4; 95%CI: 2.6, 11.3), lower family economic status (AOR = 1.9; 95%CI: 1.0, 3.4) |
| Gali N, 2017 [44] | To assess emerging nutritional problems and their association with dietary intake among school adolescents | Cross-sectional study | Urban | 546 | Mean age 15.4 (SD 1.9) | M/F | - | DDS | Mean DDS was 6.97±1.15. Cereal based diets (99.6%) and vegetables (73.9%) were the two most common foods types consumed by adolescents. |

*(Continued)*

**Table 3.** (Continued)

| Author (y) | Objective (s) | Study design | Settings | Sample | Age | Sex | Exposure | Outcomes | Main finding including description for each article |
|---|---|---|---|---|---|---|---|---|---|
| Birru SM, 2018 [84] | To assess the dietary diversity of school adolescent girls in the context of urban Northwest Ethiopia | Cross-sectional study | Urban | 768 | 10–19 | F | School type, family occupation | DDS | Adequate DDS 75.4% (95%CI (72.3, 78.6). Adequate DDS associated with attending private school (AOR = 3.2; 95%CI: 1.9,5.3), being from merchant family (AOR = 2.4; 95%CI: 1.1,5.5) |
| Seyoum Y, 2019 [72] | To assess the prevalence of adequate dietary diversity among adolescents | Cross-sectional study | Rural | 257 | 15–19 | F | - | DDS | Only 4.3% of the adolescent girls had adequate dietary diversity (WDDS ≥5) |
| Tariku A, 2019 [82] | To assess the prevalence and associated factors of dietary diversity in adolescent girls | Cross-sectional study | Urban/rural | 1550 | 10–19 | F | food insecurity | DDS | Adequate dietary diversity was 14.5 (95% CI 12.9, 16.2), Households food security 74.9% Food secure adolescent are more likely to have adequate DDS (AOR = 1.5, 95% CI 1.03, 2.1) |
| Regasa RT 2019 [80] | To determine the dietary diversity of adolescents | Cross sectional study | Urban/rural | 448 | 10–19 | F | - | DDS | Mean DDS 3.3 + 1.2 Low DDS 56%, moderate DDS 41% and high DDS 3% |

adolescents were food insecure during each consecutive round of the survey respectively [27]. In addition, 5.5% girls and 4.4% boys (P = 0.331) were from food insecure households in all three follow ups [25]. The mean height of food insecure girls was shorter by 0.87 cm (P<0.001) compared with food secure girls at baseline [25]. Predictors of food insecurity in the longitudinal study include, urban households within low (AOR = 1.7; 95% CI: 1.2, 2.5) and middle (AOR = 1.8; 95% CI: 1.2, 2.6) compared to high income tertiles were nearly twice as likely to suffer from chronic food insecurity [24]. Female sex (AOR = 1.6; 95% CI: 1.2, 2.1), high dependency ratio (AOR = 1.5; 95% CI: 1.0, 2.2) and household food insecurity (AOR = 2.7; 95% CI: 2.0, 3.6) among adolescents in urban, semi-urban, and rural areas were positively associated with food insecurity, while higher educational status was negatively associated (AOR = 0.5; 95% CI: 0.3, 0.8) [24]. Food insecure adolescents had lower DDS (P = 0.001), low mean food variety score (P = 0.001) and a lower frequency of consuming animal source foods (P = 0.001) compared to food secure adolescents [23] (Table 4).

## Eating disorders

Two studies [87, 88], both from Addis Ababa, assessed disordered eating and unhealthy weight control behaviors in adolescents [87, 88]. The prevalence of eating disorders was 8.6% (95% CI 4.9, 12.3) [87]. Female sex (AOR = 1.8; 95% CI: 1.0, 3.0) and being from less educated mother predicted a higher risk of eating disorders. Compared with no maternal schooling, maternal primary level education was associated with an AOR of 0.3 (95% CI: 0.1, 0.8), certificate/diploma with an AOR of 0.2 (95% CI: 0.1, 0.6) and a university degree or above with an AOR of 0.2 (95% CI: 0.1, 0.4) [87]. The prevalence of unhealthy weight control behavior was 31%, specifically purging behavior was 1.5% and non-purging weight control behavior was 30% [88]. In this study, predictors of unhealthy weight control behavior were being adolescent from a wealthier family (medium wealth index: AOR = 1.99; 95% CI:1.15, 3.45) and higher wealth index: AOR = 2.07; 95% CI: 1.30, 2.8), high perceived body weight (AOR = 3.01; 95%

**Table 4. Studies focusing on food insecurity.**

| First author, y | Main objective (s) | Study design | Setting: Rural/ urban | Sample size | Age (y) | Sex | Exposure (s) | Outcome (s) | Main findings |
|---|---|---|---|---|---|---|---|---|---|
| Hadley C, 2008 [26] | To examine the relationship between household and individual level food insecurity and health status among adolescent boys and girls | Data from the Jimma population based Longitudinal Family Survey of Youth (JLFSY) using multi-stage stratified cluster sampling method | Urban/ rural | 2084 | 13–17 | M/ F | Sex | Food insecurity assessed by a 6-item household food insecurity scale | Overall food insecure adolescents 13.9% (adolescents from medium food insecure households 30.8% (Boys 12.4%, girls 15.3%) and from severe food insecure household 20.5% (boys 20.9%, girls 41.0%) Boys and girls were equally likely to be living in severely food insecure households. Despite no differences in their households' food insecurity status, girls were more likely than boys to report being food insecure themselves |
| Belachew T, 2012 [24] | To identify predictors of food insecurity among adolescents | Data from the Jimma Longitudinal Family Survey of Youth (JLFSY) | Urban/ rural | 1911 | 13–17 | M/ F | Residence, | Chronic Food insecurity | 20.5% of adolescents were food insecure in the first-round survey and increased to 48.4% one year later. In the one year follow up 54.8% and 14.0% of the youth encountered transient and chronic food insecurity respectively. In urban households with low (AOR = 1.7; 95% CI: 1.2, 2.5) and middle (AOR = 1.8, 95% CI: 1.2, 2.6) income tertiles were nearly twice as likely to suffer from chronic food insecurity. Female sex (AOR = 1.6, 95% CI: 1.2, 2.1), high dependency ratio (AOR = 1.5; 95% CI: 1.0, 2.2) and household food insecurity (AOR = 2.7; 95% CI: 2.0, 3.6) were predictors of chronic adolescent food insecurity in urban, semi-urban, and rural areas, educational status of the adolescents was negatively associated with chronic food insecurity (AOR = 0.5; 95% CI: 0.3, 0.8) |

(*Continued*)

**Table 4.** (Continued)

| First author, y | Main objective (s) | Study design | Setting: Rural/ urban | Sample size | Age (y) | Sex | Exposure (s) | Outcome (s) | Main findings |
|---|---|---|---|---|---|---|---|---|---|
| Belachew T, 2013 [25] | To examine the association between food insecurity and linear growth among adolescents | The Jimma Longitudinal Family Survey of Youth (JLFSY) | Urban/ rural | 2084 | 13–17 | M/ F | Sex | Food insecurity | Food insecurity: at baseline and 1 year follow up was 15.9% in girls and 12.2% in boys (P = 0.018). In all the 3 follow-ups 5.5% girls and 4.4% boys (P = 0.331) were from food insecure households. Girls (40%) and boys (36.6%) (P = 0.045) were food insecure at least in one of the three survey rounds. Trends of food insecurity increased from 20.5% at the baseline to 48.4% on the 1 year follow up, and reversed down to 27.1% at the 2 years follow up survey. The mean height of food insecure girls was shorter by 0.87 cm (P<0.001) compared with food secure girls at baseline. But, at the follow up period, the heights of food insecure girls increased by 0.38 cm more per year compared with food secure girls (P<0.066). For boys, no significant difference in the mean height between food insecure and secured boys at baseline as well as over the follow up period. |
| Belachew T, 2013 [23] | To determine the association between adolescent food insecurity and dietary practices | Data from the first round survey the Jimma Longitudinal Family Survey of Youth (JLFSY) | Urban/ rural | 2084 | 13–17 | M/ F | Food insecurity | Dietary practice | Transient food insecure adolescents 20.5%. Coping to food insecurity: reducing daily food frequency (89.3%), worrying about running out of food (81.8%), spending the whole day without eating (23.8%) and asking for food or money to buy food/ begging (20.8%). Food insecure adolescents had low dietary diversity score (P,0.001), low mean food variety score (P,0.001) and low frequency of consuming animal source foods (P,0.001). |

*(Continued)*

**Table 4.** (Continued)

| First author, y | Main objective (s) | Study design | Setting: Rural/ urban | Sample size | Age (y) | Sex | Exposure (s) | Outcome (s) | Main findings |
|---|---|---|---|---|---|---|---|---|---|
| Mulusew G, 2017 [27] | To examine the effect of food insecurity on self-rated health status | The Jimma Longitudinal Family Survey of Youth (JLFSY) | Urban/ rural | 1,919 | 14– 22 | M/ F | - | Food insecurity assessed by 4-item adolescent food insecurity scale | 20.4%, 48.4% and 20.6% of adolescents were food insecure during each consecutive round of the survey respectively. Adolescents with food insecurity were associated with self-rated health status ($\beta = 0.28$, $P < 0.001$) |
| Juju DB, 2018 [46] | To assess food security of adolescents in the selected khat and coffee-growing areas | Cross-sectional study | Rural | 234 | 12– 18 | M/ F | Sex | Food insecurity experiences | Adolescent with food insecurity 38.0% (boys 40.2% and girls 35.9%; p-value 0.412). |
| Gizaw G, 2018 [86] | To assess the prevalence and factors associated with adolescent food insecurity among coffee producing districts of Jimma Zone | Cross-sectional study | Urban/ rural | 550 | 10– 19 | M/ F | Sex, household food insecurity, dependency ration, household head education, owner of farming land, wealth index | Adolescent food insecurity | Food insecure adolescents 59.6%. Female adolescents (AOR = 2.2; 95% CI: 1.4– 3.5), household food insecurity (AOR = 9.4; 95% CI: 5.5–16.2), male of household heads (AOR = 2.8; 95% CI: 1.4– 5.3), high dependency ratio (AOR = 2.5; 95% CI: 1.5– 4.5), not formally educated household head (AOR = 4.9; 95% CI: 2.6– 9.2) and have no own land for farm (AOR = 2.5; 95% CI: 1.2–4.9) were positively independent predictors of adolescent food insecurity. |

CI: 1.11, 8.11), higher BMI/overweight (AOR = 3.28; 95% CI:1.54, 7.01), and adolescent with severe depression (AOR = 4.09; 95% CI: 1.73,9.96) [88].

## Discussion

In this review, it was possible to extract, synthesize and summarize considerable data on nutritional status and associated factors, food insecurity, dietary diversity, micronutrient status, and disordered eating from studies among adolescents in Ethiopia. The review generally showed that there is more undernutrition (stunting, thinness and micronutrient deficiencies) than overweight among adolescents. The prevalence of thinness and stunting is higher among boys and rural adolescents whereas overweight and obesity are higher among girls and urban adolescents. The review also revealed that adolescent food insecurity and low dietary diversity are common. Consequently, a large proportion of adolescents have one or more micronutrient deficiencies. About 80% and 60% of adolescents from rural and urban settings respectively were found to have low dietary diversity. Our review supports a report from WHO [108] which documented that the magnitude of undernutrition, micronutrient deficiency, overnutrition, inadequate or unhealthy diet and life styles is high among adolescents in LMICs. The finding from the current review showed that the magnitude of undernutrition and low

DDS is substantial. Although the prevalence of overweight is low compared to that of under-nutrition, it appears that problems of overnutrition are emerging before Ethiopia has dealt with the burden of under-nutrition. This is in line with global data which shows that a double burden of malnutrition is increasing in LMICs [109] as they experience rapid economic growth, urbanization, and changes in dietary habits and levels of physical activity.

Undernutrition (underweight, stunting and thinness) is more prevalent in younger adolescents, boys, and rural adolescents, whereas overnutrition (overweight and obesity) is higher in females and urban adolescents. Adolescents in rural settings are more likely to be engaged in various labour intensive (energy consuming) domestic activities to support their family. In addition, household food insecurity is higher in rural compared to urban communities because of low literacy rates, recurrent droughts, and lack of diversity in sources of income. In contrast, because of urbanization and concomitant changes in lifestyle, urban adolescents are more likely to consume low quality foods such as sweets and fast foods, have more screen time and spend more time sedentary. There are more limited opportunities for physical activity in urban environments, especially for girls, because of overcrowding and lack of space. In a recent qualitative study, we identified that boys have more opportunity for leisure time and outdoor physical activity than girls [110].

Geographically, undernutrition is higher in northern compared to southern Ethiopia. The community in the north Ethiopia is characterized by subsistence farming where crops are the main source of income, there is greater food insecurity, and nutritional habits and experience are greatly influenced by cultural values [111] such as fasting (no animal-source meals for the majority of months of the year) [112]. In contrast, the southern region of the country is known for highly-productive horticulture of fruits and vegetables in addition to other crops, which are easily accessible to the local community.

Trends in the nutritional status of adolescents over the study period showed no clear secular trends. This could happen for the fact that the reviewed studies covered quite a limited time period, and importantly were not truly longitudinal (they represent separate studies in different populations) and are therefore not ideal for a trend analysis. The prevalence of undernutrition and overnutrition has changed little, and both have coexisted in the community over the last decade. This could happen because, despite rapid economic growth and urbanization, wide wealth disparity persists in Ethiopia. The United Nations have adopted the first ever UN Decade of Action on Nutrition, from 2016–2025 to realize the goal set to eliminate all forms of malnutrition by 2030 [113]. To date, several of the nutrition targets which were agreed upon remain unmet and on the contrary, the double burden of malnutrition challenge is increasing. It is predicted that, if current trends continue, the absolute number of overweight people will have increased from almost 2 billion today, to 3.3 billion by 2030, equal to one third of the projected world population [114]. Nutrition interventions for the current generation of adolescents in Ethiopia would require context- and community-specific intervention approaches to address all forms of malnutrition.

Micronutrient deficiencies are also common in adolescents, with deficiencies of iron, zinc, iodine, folic acid, and vitamins A and B12 being the most common. Factors that could contribute are a lack of dietary diversity, a lack of fortified foods, food insecurity and low general knowledge and awareness about the need for micronutrients for health. While there was a steady reduction in iron deficiency anemia in girls between 2000 and 2016, there is an increase in boys over the same time period. This can be explained by the targeting of national initiatives selectively towards women of reproductive age over recent decades. Despite the high burden, there are no national or regional initiatives to tackle micronutrient deficiencies in the adolescent population at ground level.

Risk factors for undernutrition identified in this review include low socioeconomic status, maternal education and dietary diversity, food insecurity, higher family size, attending a public school, younger age, male sex and living in a rural setting. Risk factors for overnutrition included female sex, urban settings, lower levels of physical activity or more sedentary life-styles, and coming from more wealthy families, having access to sweets/fast foods, older age and attending private schools. The sociodemographic and economic factors are modifiable causes of malnutrition, which could be addressed through effective context-relevant interventions, designed with the involvement of policy makers, experts, adolescents and their families.

The impact from the double burden of malnutrition could occur at the level of individual, household or nation. Individuals who were under-nourished as infants can have increased weight gain and obesity during adolescence or late in adulthood, while it is also possible for an obese person to have micronutrient deficiencies concomitantly. In the same household, some family members may be under-nourished while others are obese. The situation is the same for a given country.

Effective intervention strategies are required to tackle the double burden of malnutrition emerging in Ethiopia. The national strategy for adolescent and youth health and nutrition [115], produced by the ministry of health, recommends promoting participation and leader-ship by adolescents in the planning and implementations of adolescent-related nutrition pro-grammes, implementing innovative health education and prevention programmes using the health extension programmes, schools, mass media and digital technologies. Specifically rec-ommended interventions [115] include improving consumption of a balanced diet, with an emphasis on locally available and iron-rich foods, promoting healthy dietary habits, creating awareness of the intergenerational effects of malnutrition, creating community awareness on gender bias in household food distribution, targeted supplementation of iron and folic acid, the scaling up of facility-based nutrition assessment and counselling programs, advocacy and promotion of food fortification. These recommendations are in line with the WHO guide for implementation of effective action for improve adolescent nutrition [116]. These efforts will be more effective if global co-ordination, collaboration and integration can be achieved.

As adolescents are open for new ideas, and are concerned and interested about their health and life perspective, they could serve as the agents for change. Adolescence is therefore a win-dow of opportunity for intervention [11, 117]. Habits and experiences built during adolescence are more likely to last throughout life to some extent. Engaging adolescents in the design of their own nutrition and health interventions is likely to influence them positively. Involving young people as educators and intervention providers enables them to take responsibility for their nutritional health and is a way of allowing research to reach wider and hard-to-reach communities. A comprehensive intervention model that considers health, nutrition and well-being in general is more acceptable and impactful than targeting a single problem [116]. Such intervention models could combine counseling for nutrition and wellbeing, family life educa-tion, life skill trainings and positive behavior promotion (rather than focusing on discouraging negative behavior) to empower young people [118].

## Strengths and limitations

Strengths of this review included a rigorous, standardised methodological approach and the involvement of multidisciplinary expertise through the TALENT collaboration. We have used definition of BMI for age z-score >1 for overweight and BMI for age z-score>2 for obesity in the meta-analysis for overnutrition. A limitation was that we were not able to use data for over-weight when it was defined by weight for age z-score. Trend analysis overtime was not possible because of the limited range of years covered by the studies and the studies are mostly separate

surveys in different populations rather than longitudinal data in the same population or setting.

## Conclusions

While the magnitude of undernutrition remains high in Ethiopia, overnutrition is an emerging problem, leading to a double burden of malnutrition. Stunting and thinness are higher in boys and in rural settings while overweight and obesity are higher in girls and in urban settings. Half of adolescents found to have at least one micronutrient deficiency. There is a paucity of evidence from intervention studies to improve adolescent health and nutrition in Ethiopia. Therefore, appropriate and context-relevant intervention studies that address the various forms of malnutrition among adolescents should be designed and implemented, preferably with the active participation of adolescents themselves.

## Supporting information

**S1 Checklist.**
(DOCX)

**S1 Data.**
(XLSX)

**S2 Data.**
(XLSX)

**S3 Data.**
(XLSX)

## Acknowledgments

The researchers would like to acknowledge Jimma University for facilitating local arrangements to the study and the TALENT collaborative network for adolescent nutrition and health in sub-Saharan Africa and India for technical support.

## Author Contributions

**Conceptualization:** Mubarek Abera, Desta Hiko, Abraham Haileamlak, Caroline Fall.

**Data curation:** Mubarek Abera, Abdulhalik Workicho, Rahma Ali, Beakal Zinab.

**Formal analysis:** Mubarek Abera.

**Funding acquisition:** Mubarek Abera, Abraham Haileamlak, Caroline Fall.

**Investigation:** Mubarek Abera.

**Methodology:** Mubarek Abera, Abdulhalik Workicho, Desta Hiko, Caroline Fall.

**Project administration:** Mubarek Abera, Abraham Haileamlak, Caroline Fall.

**Resources:** Mubarek Abera, Abraham Haileamlak, Caroline Fall.

**Supervision:** Mubarek Abera, Abdulhalik Workicho, Abraham Haileamlak, Caroline Fall.

**Validation:** Mubarek Abera.

**Visualization:** Mubarek Abera, Abdulhalik Workicho, Caroline Fall.

**Writing – original draft:** Mubarek Abera.

**Writing – review & editing:** Mubarek Abera, Abdulhalik Workicho, Melkamu Berhane, Desta Hiko, Rahma Ali, Beakal Zinab, Abraham Haileamlak, Caroline Fall.

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
