## [Decision Letter · Decision Letter 0]

24 Oct 2022

PONE-D-22-20786A scoping review of adolescent nutrition in Ethiopia: transforming adolescent lives through nutrition (TALENT) initiativePLOS ONE

Dear Dr. Abera,

Thank you for submitting your manuscript to PLOS ONE. After careful consideration, we feel that it has merit but does not fully meet PLOS ONE’s publication criteria as it currently stands. Therefore, we invite you to submit a revised version of the manuscript that addresses the points raised during the review process.

We look forward to receiving your revised manuscript.

Kind regards,

Chiranjivi Adhikari, MPH, MHEd., PhD Candidate

Academic Editor

PLOS ONE

Journal Requirements:

Additional Editor Comments (if provided):

Dear authors,

It is to observe really a diligent and scientific piece of work implicating public health nutrition in a great stuff, especially at policy level, in Ethiopiya and possibly to similar but contextual neighboring countries. With most of the technical and scientific parts praiseworthy, along with a well write-up, I, would like to convey my reverence to all the reviewers for their contribution, and so, by the authors to address their comments, along with the followings:

Comments to be addressed:

1. In fig. 2-8, it is praiseworthy that trend has been shown. Additionally, it would be a good idea to synthesize and infer with trend analysis, along with intercepts and linearity/non-linearity; with their p-values. Ideas can be traced from web-based calculator: EPITOOLS (Link: https://epitools.ausvet.com.au/trend)

2. As mentioned, the quality of the studies with JBI (lines 141-3), pls include the table of the assessment for all the studies in main or supplemental file, as guideline allows.

Comments that corrections may be needed!

3. In abstract, Jamuna Bridge Institute (JBI) checklist, may need to be recheced…

4. In PRISMA chart, were the finally added 10 studies gone through eligibility? As they have been found directly included (skipping other steps?) may be the figure re-adjusted

With regards,

AE

Reviewers' comments:

Reviewer's Responses to Questions

**Comments to the Author**

1. Is the manuscript technically sound, and do the data support the conclusions?

Reviewer #1: Yes

Reviewer #2: Yes

Reviewer #3: Yes

2. Has the statistical analysis been performed appropriately and rigorously? 

Reviewer #1: N/A

Reviewer #2: N/A

Reviewer #3: N/A

3. Have the authors made all data underlying the findings in their manuscript fully available?

Reviewer #1: No

Reviewer #2: Yes

Reviewer #3: No

4. Is the manuscript presented in an intelligible fashion and written in standard English?

Reviewer #1: No

Reviewer #2: Yes

Reviewer #3: Yes

5. Review Comments to the Author

Reviewer #1: Thank you for conducting a review on a less focused but important topic - adolescent health. I would like to put forward my comments on this:

1. The authors have conducted an explicit search, quality assessment of the included articles and also one of the limitations written was "unable to conduct meta-analysis", then why is it just mentioned only as a Scoping Review? Why can't this be a Systematic review?

2. In Quality Assessment section (line 142 - 143), it is mentioned - Low quality studies were excluded. If so, this can be a Selection bias. But, in Study selection section (line 152 - 153), it is mentioned - No studies were excluded. Please clarify.

3. Data availability - No (Some restrictions may apply), can you please mention what and which restrictions would apply for further clarification (as this is a review)?

4. In PRISMA diagram, spelling of Meta (between Eligibility and Screening section) is incorrect and some brackets are not closed. Please check. Also, in the last section of diagram "Included", it is mentioned "Studies included in quantitative synthesis (meta-analysis) (n=74)", but MA has not been conducted, kindly clarify.

5. If possible, title can be refined and made specific.

Reviewer #2: A nice paper on an important topic. Could just do with a few minor clarifications:

Abstract,

Line 22: diets, rather than diet

Lines 27, methods: I think ‘on adolescent nutrition’ is too vague. I appreciate that word limit is tight in the abstract but you specific ‘on prevalence and interventions for all forms of malnutrition, with no limits on study design’ ? or something similar? Otherwise it’s a slightly confusing start to the abstract. Especially since no intervention studies are mentioned in the results, you wonder if these weren’t included.

Introduction

Line 55: suggest changing ‘(overweight and obesity) is more important’ to ‘(overweight and obesity) is more prevalent’, since importance suggestions they don’t put ‘importance’ on undernutrition.

Line 58-60: “This, alongside persisting 59 undernutrition in large sections of the population results in a double burden of malnutrition in 60 LMICs, compounded by low levels of government investment to solve the problem” I would love to see a reference to this new and comprehensive review here. You may also find it helpful for the discussion section as it puts Ethiopia in the context of the rest of the region: Wrottesley SV, Mates E, Brennan E, Bijalwan V, Menezes R, Ray S, Ali Z, Yarparvar A, Sharma D, Lelijveld N. Nutritional status of school-age children and adolescents in low-and middle-income countries across seven global regions: a synthesis of scoping reviews. Public health nutrition. 2022 Feb 14:1-33.

Methods

Line 141: define acronym JBI on first use in the main text

It would be good to see some definitions of malnutrition in either the methods or the results. For example, you talk about ‘stunting’ in the results, but the definition of this isnt always standard so it would be good to know what you were defining as ‘stunting’ for this population age group. Same for the other anthropometry and the micronutrient deficiencies.

Results:

Figure 2-8 – I was avoid saying ‘trends’ but rather ‘reported prevalence’s’ because trends suggests that the assessment are linked to each other. I would also include the definition of nutritional status in the footnotes for the figures e.g. overweight and obesity based on WHO 2007 growth reference BAZ>+1

Diet diversity, lines 280-309: could you mention the tools used to assess DDS? This is always a confusing area in adolescent nutrition – which tools are validated and which tools are used, so good to mention that here. Same for tools used to assess food insecurity.

You don’t mention intervention studies, did you not find any?

Discussion:

You say “The review generally showed that there is more undernutrition (thinness and underweight)” – if you are talking about low BMI-for-age z-score, then just say thinness… if you are talking about weight-for-age z-score, say underweight. I think underweight can be removed from here. See: Lelijveld N, Benedict RK, Wrottesley SV, Bhutta ZA, Borghi E, Cole TJ, Croft T, Frongillo EA, Hayashi C, Namaste S, Sharma D. Towards standardised and valid anthropometric indicators of nutritional status in middle childhood and adolescence. The Lancet Child & Adolescent Health. 2022 Aug 24. Also, undernutrition generally includes micronutrient deficiencies too, so perhaps say “there is more undernutrition (thinness and micronutrient deficiencies)”

Define DDS when you first use it in the text

Limitations: I think the heterogeneity of studies requires a little more explanation – there are important learnings for nutritionists working with this age group around the lack of standardisation in research methods. Presumably, lots of the studies focused on different age groups within the 10-19 range? What other heterogeneity did you find? In definitions of malnutrition? Please expand

Reviewer #3: The review was good.

Abstract: Please check for Jamuna Bridge Institute or Joanna Briggis Institute..short form JBI

Introduction:

Line 45: second applies only to fetal life and infancy; remove it. 

Line 56-60: check it, Line 56-58 (remove it).

Line 77: Knowledge ..replace term with evidence or other appropirate terms

Line 78: Thus, this review aimed at understanding the nutritional statusof adolescents in Ethopia (delete it). the above line 76-78 give same meaning.

Results:

Please check line 197-98 once.

Provide reference for line 229-233.

In Line 248: Use another word for: "In the same survey"

Line 322-327: clearly rewrite

Line 332: delete "has produced 5 articles"

Eating disorder: rewrite last 4 line

Discussion: Discuss the main findings of the study with other study too (similar and contrast)

Strength and limitations: check the limitations for spelling error, also check for figure 1.

Conclusion: need to rewrite, highligting all research questions.

6. PLOS authors have the option to publish the peer review history of their article (what does this mean?). If published, this will include your full peer review and any attached files.

Reviewer #1: **Yes: **Ms. Priyanka Akshay Shah

Reviewer #2: No

Reviewer #3: **Yes: **Rojana Dhakal

---

## [Author Response · Author response to Decision Letter 0]

28 Dec 2022

Response to Feedback

Our general response: First of all, the authors would like to acknowledge the reviewer/s for taking time to read and give us important feedback/comments on our manuscript. We have considered these and are grateful for the opportunity to edit and improve the manuscript. Please find below a point-by-point response to the issues raised. We have used track change in the main document to indicate changes. 

1. In fig. 2-8, it is praiseworthy that trend has been shown. Additionally, it would be a good idea to synthesize and infer with trend analysis, along with intercepts and linearity/non-linearity; with their p-values. Ideas can be traced from web-based calculator: EPITOOLS (Link: https://epitools.ausvet.com.au/trend)

Response: We acknowledge this feedback. However we were not able to do this analysis partly because of the limited range of years covered, but also because the studies are mostly separate surveys in different populations rather than longitudinal data in the same population or setting. We have stated and given emphasis for this in Results section. 

2. As mentioned, the quality of the studies with JBI (lines 141-3), pls include the table of the assessment for all the studies in main or supplemental file, as guideline allows.

Comments that corrections may be needed!

Response: Thank you for this feedback. Now we have included the table as additional file. 

3. In abstract, Jamuna Bridge Institute (JBI) checklist, may need to be rechecked…

Response: Thank you for the feedback. Now this is corrected as “Joanna Briggs Institute” 

4. In PRISMA chart, were the finally added 10 studies gone through eligibility? As they have been found directly included (skipping other steps?) may be the figure re-adjusted

Response: This is also now revised and corrected by bringing those 10 studies up to the front (top).

5. Reviewer #1: Thank you for conducting a review on a less focused but important topic - adolescent health. I would like to put forward my comments on this:

5.1. The authors have conducted an explicit search, quality assessment of the included articles and also one of the limitations written was "unable to conduct meta-analysis", then why is it just mentioned only as a Scoping Review? Why can't this be a Systematic review?

Response: Thank you for this feedback. In the current revised version we have added a meta-analysis for stunting, thinness and overweight/obesity. As such the title is reworded by indicating the work as “systematic review and meta-analysis” 

5.2. In Quality Assessment section (line 142 - 143), it is mentioned - Low quality studies were excluded. If so, this can be a Selection bias. But, in Study selection section (line 152 - 153), it is mentioned - No studies were excluded. Please clarify.

Response: The plan to exclude low quality studies was mentioned in the Methods section. But after searching we did not found articles to be of low quality on those that full filed the inclusion criteria. We have added this clarification in the result (1st paragraph last sentence)

5.3. Data availability - No (Some restrictions may apply), can you please mention what and which restrictions would apply for further clarification (as this is a review)?

Response: Thank you for this feedback. As this is a secondary review almost all of the data is reported in the table within the main paper. As such we do not have data that is restricted with us. 

5.4. In PRISMA diagram, spelling of Meta (between Eligibility and Screening section) is incorrect and some brackets are not closed. Please check. Also, in the last section of diagram "Included", it is mentioned "Studies included in quantitative synthesis (meta-analysis) (n=74)", but MA has not been conducted, kindly clarify.

Response: Now this is also corrected as “Meta-analysis”

5.5. If possible, title can be refined and made specific.

Response: A slight amendment is now done on the title. 

6. Reviewer #2: A nice paper on an important topic. Could just do with a few minor clarifications:

6.1. Abstract: Line 22: diets, rather than diet: 

Response: Now corrected as ‘diets’

6.2. Abstract: Lines 27, methods: I think ‘on adolescent nutrition’ is too vague. I appreciate that word limit is tight in the abstract but you specific ‘on prevalence and interventions for all forms of malnutrition, with no limits on study design’ ? or something similar? Otherwise it’s a slightly confusing start to the abstract. Especially since no intervention studies are mentioned in the results, you wonder if these weren’t included.

Response: Now corrected as “A systematic search of electronic databases for published studies on the prevalence of and interventions for adolescent malnutrition in Ethiopia in the English language since the year 2000 was performed using a three-step search strategy”

Introduction

6.3. Line 55: suggest changing ‘(overweight and obesity) is more important’ to ‘(overweight and obesity) is more prevalent’, since importance suggestions they don’t put ‘importance’ on undernutrition.

Response: Now this is corrected and the term ‘important’ is replaced by ‘prevalent

6.4. Line 58-60: “This, alongside persisting 59 undernutrition in large sections of the population results in a double burden of malnutrition in 60 LMICs, compounded by low levels of government investment to solve the problem” I would love to see a reference to this new and comprehensive review here. You may also find it helpful for the discussion section as it puts Ethiopia in the context of the rest of the region: Wrottesley SV, Mates E, Brennan E, Bijalwan V, Menezes R, Ray S, Ali Z, Yarparvar A, Sharma D, Lelijveld N. Nutritional status of school-age children and adolescents in low-and middle-income countries across seven global regions: a synthesis of scoping reviews. Public health nutrition. 2022 Feb 14:1-33.

Response: thank you for this. We have now made use of the suggested article and referenced it as well. 

Methods

6.5. Line 141: define acronym JBI on first use in the main text

Response: corrected as ‘Joanna Briggs Institute (JBI)’

6.6. It would be good to see some definitions of malnutrition in either the methods or the results. For example, you talk about ‘stunting’ in the results, but the definition of this isn’t always standard so it would be good to know what you were defining as ‘stunting’ for this population age group. Same for the other anthropometry and the micronutrient deficiencies.

Response: Thank you for this feedback. Now we have added this definitions in the outcome section. 

6.7. Results:Figure 2-8 – I was avoid saying ‘trends’ but rather ‘reported prevalence’s’ because trends suggests that the assessment are linked to each other. 

Response: We are thankful for this feedback. However rather than directly removing this term we have added in the result section that the findings of the different studies are not linked to each other and rather are extracted from different independent studies done in different settings. We also have highlighted this in the discussion section.

6.8. I would also include the definition of nutritional status in the footnotes for the figures e.g. overweight and obesity based on WHO 2007 growth reference BAZ>+1

Response: We accepted and included the suggested comment. “Correction done: Stunting (Height-for-age z-score <-2) Thinness (BMI-for-age z-score <-2), underweight (weight-for-age z-score <-2), overweight (Body mass index-for-height z-score is >1), Obesity (weight-for-height z-score is > 2)”

6.9. Diet diversity, lines 280-309: could you mention the tools used to assess DDS? This is always a confusing area in adolescent nutrition – which tools are validated and which tools are used, so good to mention that here. Same for tools used to assess food insecurity.

Response: Comment accepted and corrected as DDS measured using the Food and Agriculture Organization of the United Nations (FAO), and food insecurity using the Household Food Insecurity Access Scale (HFIAS). 

6.10. You don’t mention intervention studies, did you not find any?

Response: Thank you for this feedback. Yes there are no many interventional studies on adolescent nutrition. But we have included the available ones such as (Tamiru D, 2016 (28)) in this study because of the lack of well designed and implemented interventional studies for improving adolescent nutrition, we made a focus on synthesizing burden of the problem. 

6.11. Discussion: You say “The review generally showed that there is more undernutrition (thinness and underweight)” – if you are talking about low BMI-for-age z-score, then just say thinness… if you are talking about weight-for-age z-score, say underweight. I think underweight can be removed from here. See: Lelijveld N, Benedict RK, Wrottesley SV, Bhutta ZA, Borghi E, Cole TJ, Croft T, Frongillo EA, Hayashi C, Namaste S, Sharma D. Towards standardised and valid anthropometric indicators of nutritional status in middle childhood and adolescence. The Lancet Child & Adolescent Health. 2022 Aug 24. Also, undernutrition generally includes micronutrient deficiencies too, so perhaps say “there is more undernutrition (thinness and micronutrient deficiencies)”

Response: Comment accepted and corrected in the discussion section. 

6.12. Define DDS when you first use it in the text

Response: Now this is corrected as “diet diversity score (DDS)”

6.13. Limitations: I think the heterogeneity of studies requires a little more explanation – there are important learnings for nutritionists working with this age group around the lack of standardisation in research methods. Presumably, lots of the studies focused on different age groups within the 10-19 range? What other heterogeneity did you find? In definitions of malnutrition? Please expand

Response: thank you for this feedback too. Now we have added this in the discussion section specifically for underweight and overweight. . 

7. Reviewer #3: The review was good.

Abstract: Please check for Jamuna Bridge Institute or Joanna Briggis Institute..short form JBI

Response: thank you for this feedback. Now we corrected as ‘Joanna Briggs Institute (JBI)

7.1. Introduction: Line 45: second applies only to fetal life and infancy; remove it. 

Response: We are thankful for this feedback. However, we would like to retain this, because our intention is to indicate that adolescence is a second window of opportunity for investment on nutrition and growth, for long-term benefit. We think this is an important concept for policy makers and decision makers and planners that the opportunity for nutrition intervention is not over during early childhood. There is also another opportunity for such investment, which is during adolescence. 

7.2. Line 56-60: check it, Line 56-58 (remove it).

Response: Thank you for this comment. Again rather than removing the suggested section we kept as it was because our aim is to make a contrast adolescent malnutrition as a public health problem in the previous and current time. 

7.3. Line 77: Knowledge.replace term with evidence or other appropirate terms

Response: comment accepted and the “knowledge’ is replace by “evidence” 

7.4. Line 78: Thus, this review aimed at understanding the nutritional status of adolescents in Ethopia (delete it). the above line 76-78 give same meaning.

Response: Comment accepted and the statement is deleted now. 

Result 

7.5. Please check line 197-98 once.

Response: Thank you for this feedback. Now we understand the original statement was less clear. Now we have revised and edited that statement as “In all of the five studies that have reported overnutrition, the prevalence of overweight/obesity was higher in girls (28,30,40,42–44,49,52), and in three of four studies that have reported under nutrition, the prevalence of underweight was higher in boys (28,30,35). 

7.6. Provide reference for line 229-233”

Response: Now references are indicated (21,29,31,32,38,40,50,58,61,63).

7.7. In Line 248: Use another word for: "In the same survey"

Response: now the phrase "In the same survey" is replaced by “Based on the micronutrient survey,

7.8. Line 322-327: clearly rewrite

Response: comment is accepted and the sentence “The prevalence of food insecurity in Jimma zone (urban and rural settings) was 59.6%.

7.9. Line 332: delete "has produced 5 articles"

Response: Comment is accepted and the phrase is now removed/deleted. 

7.10. Eating disorder: rewrite last 4 line

Response: This statement is also corrected and clarified as “In this study, predictors of unhealthy weight control behavior were being adolescent from a wealthier family (medium wealth index : AOR=1.99; 95% CI:1.15, 3.45) and higher wealth index : AOR=2.07; 95% CI: 1.30, 2.8), high perceived body weight (AOR= 3.01; 95% CI: 1.11, 8.11), higher BMI/overweight (AOR= 3.28; 95% CI:1.54, 7.01), and adolescent with severe depression (AOR= 4.09; 95% CI: 1.73,9.96)”

7.11. Discussion: Discuss the main findings of the study with other study too (similar and contrast)

7.12. Strength and limitations: check the limitations for spelling error, also check for figure 1.

Response: in the limitation section, “A a limitation” is corrected as “A limitation”. Also “fig. 1” is corrected as “Figure 1”.

7.13. Conclusion: need to rewrite, highligting all research questions.

Response: We have corrected this by adding statement “Stunting and thinness are higher in boys and in rural settings while overweight and obesity are higher in girls and in urban settings. Half of adolescents found to have at least one micronutrient deficiency”

---

## [Editor Report · Decision Letter 1]

8 Jan 2023

A systematic review and meta-analysis of adolescent nutrition in Ethiopia: transforming adolescent lives through nutrition (TALENT) initiative

PONE-D-22-20786R1

Dear Dr. Abera,

We’re pleased to inform you that your manuscript has been judged scientifically suitable for publication and will be formally accepted for publication once it meets all outstanding technical requirements.

Kind regards,

Chiranjivi Adhikari, MPH, MHEd., PhD Candidate

Academic Editor

PLOS ONE

Additional Editor Comments (optional):

Dear author(s),

Thank you for the updated version of the manuscript. Now, although it is suitable for acceptance, I still find "Jamuna Bridge Institute" in Abstract, in line 28. So, It needs further rigorous grammar, syntax and other checks at least once by all the authors, after keeping up the formats as per journal guideline.

So, I recommend to submit a final version.

With regards,

Chiranjivi, AE

Reviewers' comments:

<quillbot-extension-portal></quillbot-extension-portal>

---

## [Editor Report · Acceptance letter]

13 Jan 2023

PONE-D-22-20786R1 

A systematic review and meta-analysis of adolescent nutrition in Ethiopia: transforming adolescent lives through nutrition (TALENT) initiative 

Dear Dr. Abera:

I'm pleased to inform you that your manuscript has been deemed suitable for publication in PLOS ONE. Congratulations! Your manuscript is now with our production department. 

Kind regards, 

on behalf of

Mr. Chiranjivi Adhikari 

Academic Editor

PLOS ONE